# NarrLV: Towards a Comprehensive Narrative-Centric Evaluation for Long Video Generation

**Xiaokun Feng**[1,2,3,*]   **Haiming Yu**[3]   **Meiqi Wu**[3,4]   **Shiyu Hu**[5]   **Jintao Chen**[3,6]
**Chen Zhu**[3]   **Jiahong Wu**[3‡]   **Xiangxiang Chu**[3]   **Kaiqi Huang**[1,2†]
[1] School of Artificial Intelligence, UCAS [2] CASIA [3] AMAP, Alibaba Group
[4] School of Computer Science and Technology, UCAS
[5] School of Physical and Mathematical Sciences, NTU [6] PKU
**Project Page:** https://amap-ml.github.io/NarrLV-Website/

## Abstract

With the rapid development of foundation video generation technologies, long video generation models have exhibited promising research potential thanks to expanded content creation space. Recent studies reveal that the goal of long video generation tasks is not only to extend video duration but also to accurately express richer narrative content within longer videos. However, due to the lack of evaluation benchmarks specifically designed for long video generation models, the current assessment of these models primarily relies on benchmarks with simple narrative prompts (*e.g.*, VBench). To the best of our knowledge, our proposed **NarrLV** is the first benchmark to comprehensively evaluate the **Narr**ative expression capabilities of **L**ong **V**ideo generation models. Inspired by film narrative theory, (**i**) we first introduce the basic narrative unit maintaining continuous visual presentation in videos as Temporal Narrative Atom (TNA), and use its count to quantitatively measure narrative richness. Guided by three key film narrative elements influencing TNA changes, we construct an automatic prompt generation pipeline capable of producing evaluation prompts with a flexibly expandable number of TNAs. (**ii**) Then, based on the three progressive levels of narrative content expression, we design an effective evaluation metric using the MLLM-based question generation and answering framework. (**iii**) Finally, we conduct extensive evaluations on existing long video generation models and the foundation generation models. Experimental results demonstrate that our metric aligns closely with human judgments. The derived evaluation outcomes reveal the detailed capability boundaries of current video generation models in narrative content expression.

## 1 Introduction

Video generation has consistently been regarded as a long-term research goal (Xing et al., 2024), from the earliest techniques with subtle motion effects (Vondrick et al., 2016) to recent foundation models like Wan (Wang et al., 2025) that achieve high-fidelity dynamic video generation. Given that these models are limited to producing short videos, recent studies have shifted focus toward designing long video generation models (Xing et al., 2024). Benefiting from a broader content creation space, long video generation models (Waseem & Shahzad, 2024) show greater potential to meet practical needs in areas such as film production and world simulation (Cho et al., 2024; Mao et al., 2025).

Some approaches (Kim et al., 2024; Zhao et al., 2025) have incorporated innovative designs into denoising models, enabling foundation video generation models (Chen et al., 2024a; Yang et al., 2024b) to produce more frames. However, the goal of long video generation goes beyond extending video duration. It critically involves accurately and appropriately conveying richer narrative content in extended videos (Waseem & Shahzad, 2024; Bansal et al., 2024a). Existing long video models often focus on leveraging temporally evolving narrative texts to guide video generation across different time segments, thereby enhancing the narrative content in the generated videos. Models such as FreeNoise (Qiu et al., 2024), Presto (Yan et al., 2024), and Mask²DiT (Qi et al., 2025) emphasize

---

* Work done during the internship at AMAP, Alibaba Group.     † Corresponding author     ‡ Project lead

efficient interaction between segmented texts with diverse narrative semantics and corresponding video clip features, reflecting the field's pursuit of generating narrative-rich long-duration videos.

Unlike the rapid development of long video generation models, the evaluation benchmarks for this task appear somewhat lagging. Early models like NUWA-XL (Yin et al., 2023), Loong (Wang et al., 2024b), and FreeNoise (Qiu et al., 2024) used conventional metrics (FID (Heusel et al., 2017), FVD (Unterthiner et al., 2019), CLIP-SIM (Radford et al., 2021)), which are often misaligned with human judgment (Otani et al., 2023). To address this, numerous benchmarks (Huang et al., 2024a; Liu et al., 2024b; Ling et al., 2025; Chen et al., 2025e) for video generation have been proposed, yet there is still a lack of benchmarks specifically designed for long video generation. This leads to recent models, such as Prestro (Yan et al., 2024), GLC-Diffusion (Ma et al., 2025), and SynCoS (Kim et al., 2025), typically being evaluated on a general benchmark, VBench (Huang et al., 2024a). Although VBench encompasses a wide range of evaluation dimensions, its prompts generally consist of brief narratives, limiting its effectiveness in assessing the models' ability to convey rich narrative content.

To evaluate the **Narr**ative expression capabilities of **L**ong **V**ideo generation models, we propose a novel benchmark, **NarrLV**, inspired by film narrative theory (Verstraten, 2009). Firstly, to quantify the abstract concept of narrative content richness, we define the smallest narrative unit maintaining continuous visual presentation as a Temporal Narrative Atom (**TNA**). The number of TNAs serves as a quantitative measure of narrative richness, as illustrated by the prompts shown in Fig. 1 (a). Fig. 1 (b) shows that representative benchmarks (Huang et al., 2024a; Wang et al., 2024a; Feng et al., 2024a) concentrate on prompts with only a small number of TNAs in a narrow range (please see App. A.1 for details), which limits their evaluation to simple narratives with limited richness. To thoroughly assess the full narrative capabilities of long video generation models, we construct an innovative prompt suite that can flexibly expand narrative content richness. Specifically, based on the 6D principles of film narratology (Cutting, 2016; Hu et al., 2023a), we identify three key dimensions affecting TNA changes: scene attributes, object attributes, and object actions. Subsequently, we use the Large Language Model (LLM) (Yang et al., 2024a) to establish an automatic prompt generation pipeline capable of generating test prompts that cover a wide range of TNA numbers.

Corresponding to the prompt suite focused on narrative content, we design an effective evaluation metric following a progressive narrative expression paradigm (Chatman & Chatman, 1980; Roberts et al., 1996; Cowie, 2013). From the basic elements of scenes and objects to the narrative units they form, our metric encompasses three evaluative dimensions: narrative element fidelity, narrative unit coverage, and narrative unit coherence. Considering the flexible and diverse nature of narrative content, our implementation leverages the MLLM-based (Bai et al., 2025; Hurst et al., 2024) question generation and answering framework (Hu et al., 2023b; Yarom et al., 2023; Cho et al., 2023a) , which can create extensible question sets according to varying TNA numbers. Finally, we conduct comprehensive evaluations of existing long video generation models (Bansal et al., 2024b; Kim et al., 2024; Qiu et al., 2024; Lu et al., 2024; Zhao et al., 2025) and the foundation models (Wang et al., 2025; Kong et al., 2024; Yang et al., 2024b; Zheng et al., 2024b; Lin et al., 2024) they are often built upon. The experimental results show that our metrics align well with human preferences and provide detailed insights into the narrative expression boundaries of current models.

Our key contributions are as follows: **(i)** In light of the lack of benchmarks for long video generation models, we propose NarrLV, a novel benchmark focusing on narrative content expression capabilities. **(ii)** Inspired by film narrative theory, NarrLV comprises a thorough prompt suite with flexibly expandable narrative content, and an effective evaluation metric based on progressive narrative expression. **(iii)** We conduct comprehensive evaluations of existing long video and foundation generation models using our metrics, which demonstrate high alignment with human preferences.

## 2 RELATED WORKS

**Long video generation models.** Owing to high computational costs in video feature processing (Kim et al., 2024), foundation video generators (*e.g.*, CogVideoX (Yang et al., 2024b) and Wan (Wang et al., 2025)) typically produce short videos. Comparatively, long video generation models generally refer to those capable of generating longer videos than these foundation models (Zhao et al., 2025; Kim et al., 2024; Lu et al., 2024). In practice, most long video models are extensions of short-video foundation models. FreeLong (Lu et al., 2024) generates more frames by balancing the feature frequency distribution for long videos. RIFLEx (Zhao et al., 2025) achieves a $3\times$ extension

of video duration by adjusting temporal position encoding. In addition to pursuing longer video durations, long video generation tasks also focus on accurately conveying richer narrative content in extended videos. Specifically, videos generated at different time intervals need to be guided by textual narratives that evolve over time (Zhou et al., 2024; Tian et al., 2024; Bansal et al., 2024a; Qiu et al., 2024; Qi et al., 2025). These temporally changing textual descriptions form rich narrative content and pose new challenges for model design. FreeNoise (Qiu et al., 2024) progressively injects segmented texts regarding object movement evolution into different denoising steps. Presto (Yan et al., 2024) proposes an innovative segmented cross-attention strategy, directly facilitating the interaction between latent features of long videos and segmented narrative texts. Addressing the current lack of benchmarks specifically designed for long video generation models, we develop a novel benchmark, NarrLV, focused on narrative expression capabilities.

**Video generation evaluation.** The growing capabilities of video generation models continually introduce new demands for effectively evaluating the generated videos (Liu et al., 2024a). Early evaluations primarily rely on generic metrics (*e.g.*, FID (Heusel et al., 2017), IS (Salimans et al., 2016), FVD (Unterthiner et al., 2019), which often exhibit significant deviations from human perception (Huang et al., 2024a; Otani et al., 2023) and provide limited insight into model capabilities (Zheng et al., 2025). To better evaluate various model capabilities, several specialized benchmarks have been proposed. For instance, VBench (Huang et al., 2024a) defines 16 evaluation dimensions based on video quality and video-condition consistency. DEVIL (Liao et al., 2024) emphasizes video dynamism; TC-Bench (Feng et al., 2024a) evaluates temporal compositionality; and VMBench (Ling et al., 2025) thoroughly assesses motion quality. StoryEval (Wang et al., 2024a) is another related benchmark that evaluates event-level story presentation capability using prompts of 2 to 4 consecutive events. However, all these benchmarks

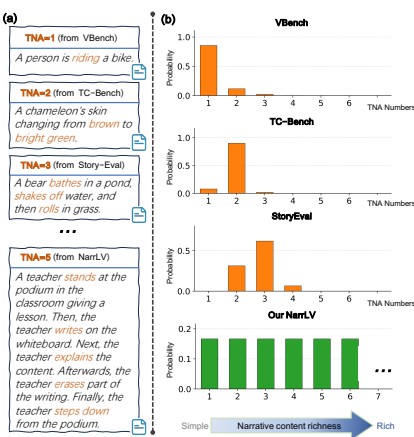

Figure 1: **(a)** Prompt examples with varying numbers of TNAs. **(b)** Comparison of TNA count distributions across different benchmark.

primarily object short-duration models. As shown in Fig. 1, their prompts contain relatively few TNAs with a narrow distribution, making them insufficient for testing models on complex, extended narrative content. In contrast, NarrLV is designed to fill this gap by providing a platform to evaluate the generative capacity of long video generators under rich and comprehensive narrative content.

## 3 NARRLV

The overall framework of our NarrLV is illustrated in Fig. 2. First, building on film narrative theory (Verstraten, 2009; Cutting, 2016), we introduce the Temporal Narrative Atom (TNA) as a unit to measure the richness of narrative content and identify three key dimensions (Hu et al., 2023a) that influence its count. Subsequently, we develop an automated prompt generation pipeline capable of producing evaluation prompts with a flexibly expandable number of TNAs. The resulting prompt suite enables comprehensive assessment of the model's generation capability across various levels of narrative content richness. Finally, leveraging the MLLM-based question generation and answering framework (Hu et al., 2023b; Yarom et al., 2023; Cho et al., 2023a), we construct a comprehensive evaluation metric founded on the three progressive levels of narrative content expression. In the following sections, we will provide detailed introductions to each component.

### 3.1 PRELIMINARIES OF FILM NARRATIVE THEORY

Film narratology (Verstraten, 2009; Kuhn, 2009) is a discipline dedicated to the study of narrative structures and expressive techniques in films. To evaluate video generation models with emphasis on narrative expression, we draw upon relevant theories from this field. First, the richness of narrative content is an abstract concept. To facilitate its quantification, it is necessary to define a basic unit for measuring narrative richness (McKee, 2005). Drawing from the definition of *Beat* in film narratology (McKee, 2005), we define the smallest narrative unit in continuous visual expression as the Temporal

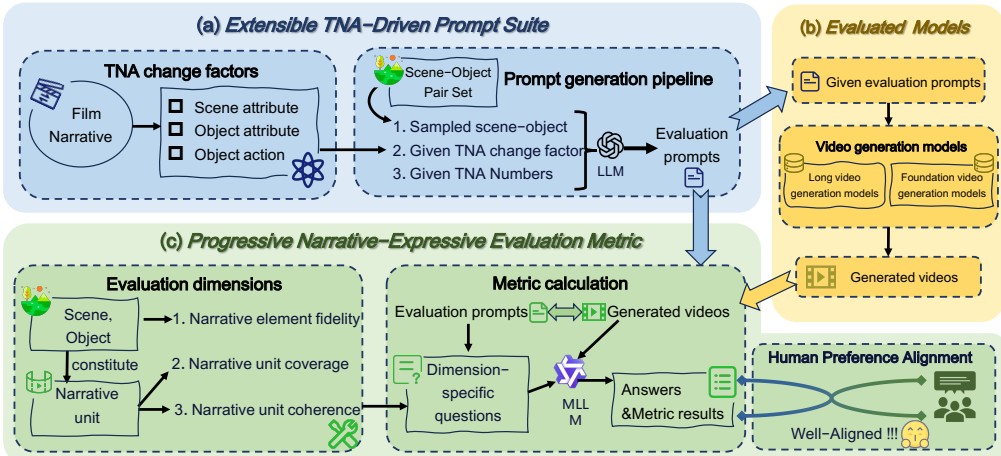

Figure 2: **Framework of our NarrLV.** **(a)** Our prompt suite is inspired by film narrative theory and identifies three key factors influencing Temporal Narrative Atom (TNA) transitions. Based on these, we construct a prompt generation pipeline capable of producing evaluation prompts with flexibly adjustable TNA counts. **(b)** Our evaluation models include long video generation models and the foundation models they often rely on. **(c)** Based on the progressive expression of narrative content, we conduct evaluations from three dimensions, employing an MLLM-based question generation and answering framework for calculations. Our metric is well-aligned with human preferences.

Narrative Atom (TNA). Fig. 1 presents several prompt examples containing different numbers of TNAs. Evidently, the greater the number of TNAs, the richer the corresponding narrative content.

Following this, a naturally arising question is: what factors influence the number of TNAs? The 6D principles of film narrative (Cutting, 2016; Hu et al., 2023a) divide the narrative content into six critical elements based on spatiotemporal and causal relationships in video: *total frame*, *temporal continuity*, *spatial continuity*, *scene*, *action*, and *object*. In the context of video generation tasks, the total frame count, *i.e.*, video length, is determined by the inherent characteristics of the generation model. Regarding temporal and spatial continuity, existing generation models typically assume a setting of continuous spatio-temporal change (Cho et al., 2024; Liu et al., 2024a). Specifically, when constructing training datasets, they explicitly exclude samples that are spatio-temporally discontinuous due to factors like shot cuts (Kong et al., 2024). Therefore, the adjustable factors that can alter the narrative richness are limited to scene, object, and action. Based on this, we identify three key variable factors influencing the number of TNAs: **scene attributes**, **object attributes**, and **object actions**, formalized as $F = [s_{\text{att}}, t_{\text{act}}, t_{\text{att}}]$. These factors are similar to the temporal composition factors mentioned in TC-Bench (Feng et al., 2024a). However, unlike TC-Bench, which primarily focuses on two TNAs, our prompt suite emphasizes the flexible extensibility of TNA count.

## 3.2 EXTENSIBLE TNA-DRIVEN PROMPT SUITE

A key feature of our benchmark is the introduction of prompts that enable flexible TNA extensibility to thoroughly assess the narrative expression capabilities of video generation models. To achieve this goal while minimizing the time-consuming and labor-intensive manual design processes (Huang et al., 2024a; Feng et al., 2024a), we develop an automatic prompt generation pipeline based on the LLM (Yang et al., 2024a). Considering that scenes and objects are the primary factors influencing TNA numbers, our pipeline first aggregates a comprehensive set of scene-object pairs. Then, we sample specific scene-object instances and utilize the LLM to generate specific test prompts by integrating their potential attribute and action evolution.

**Acquisition of scene-object pair set.** To ensure that our test prompts closely align with the video content that users typically focus on, our data source includes the recently released and user-focused dataset VideoUFO (Wang & Yang, 2025), which effectively reflects real-world applicability scenarios (Wang & Yang, 2024). Additionally, we incorporate the latest DropletVideo (Zhang et al., 2025) dataset, which features rich narrative content. Specifically, we randomly sample $100k$ text prompts from VideoUFO-1M (Wang & Yang, 2025) and DropletVideo-1M (Zhang et al., 2025), respectively. Subsequently, we employ an LLM (Yang et al., 2024a) to analyze these $200k$ prompts individually,

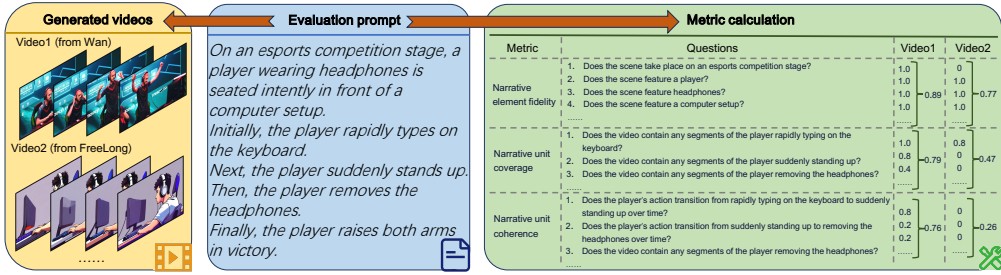

Figure 3: **Illustration of our metric evaluation process.** Given an evaluation prompt, different video generation models produce corresponding video outputs. Concurrently, based on the semantic information within the prompt, judgment questions concerning different evaluation dimensions are generated, resulting in evaluation outcomes for the generated videos. Better viewed with zoom-in.

extracting the scene $s$ and major object list $o$ corresponding to each text. Next, we merge the object lists under the same scene to obtain the final scene-object pairs $s_o$. For instance, in a *basketball court* scene, the object list includes *basketballs*, *players*, *referees*, and other related objects. Ultimately, the aggregated $s_o$ constitutes our scene-object pair set $S_O$ (see App. A.2 for for detailed implementation and statistical analysis).

**Automatic prompt generation.** As shown in the pipeline of Fig. 2 (a), we first extract a specific scene-object instance $s_o$ from $S_O$. Then, we randomly sample 1 to 2 objects from the object list in $s_o$. Next, we specify the TNA change number $n$ and the TNA change factor $f$, utilizing an LLM to incorporate the potential attribute/action evolution process. For detailed prompt instructions, please refer to App. A.3. Finally, we obtain a test prompt $p_{f,n}$ corresponding to $n$ and $f$, formalized as:

$$(s_o, f, n) \xrightarrow{\text{LLM}} p_{f,n}, \quad \text{where } s_o \in S_O, \ f \in F, \ n \in [1, N_{tna}]. \tag{1}$$

**Post-processing.** Based on the aforementioned pipeline, we can quickly generate large-scale prompts encompassing different TNA change factors and numbers. Considering the rising computational costs of video generation models (*e.g.*, Wan2.1-14B (Wang et al., 2025) requires about 110 minutes to produce a video on an H20 GPU), it is necessary to perform post-processing to carefully select a small yet representative prompt suite (Huang et al., 2024a). First, for scene-object pair set $S_O$ with large quantities, we categorize them into 14 major categories (see App. A.2 for more details). For instance, under the *sports venue* category, there are subsets for *football fields*, *basketball courts*, etc.

Under each factor $f$ and number $n$, we select 1 to 3 $s_o$ from each major category, ultimately obtaining 20 evaluation prompts $\{p_{f,n}^i\}_{i=1}^{20}$. In addition, we temporarily set the maximum TNA number $N_{tna}$ to 6, and observe that this range can already reveal some insightful conclusions (see Sec. 4.2). With 3 change factors, we evaluate the models under $20 \times 6 \times 3 = 360$ prompts. It is important to note that our prompt generation pipeline has good extensibility. For longer video generation in the future, we can follow the same process to obtain prompts with a broader TNA distribution.

### 3.3 PROGRESSIVE NARRATIVE-EXPRESSIVE EVALUATION METRIC

To systematically evaluate the narrative quality of long video generation, we introduce three core metrics—**Narrative Element Fidelity**, **Narrative Unit Coverage**, and **Narrative Unit Coherence**—grounded in audiovisual storytelling principles (Chatman & Chatman, 1980; Roberts et al., 1996; Cowie, 2013; Diniejko, 2010). These dimensions provide a rational approach for assessing narrative expression by progressively focusing on the basic elements of scenes and objects and the temporal narrative units they form.

Furthermore, given the inherently flexible and diverse nature of narrative content, traditional task-specific models (Hinz et al., 2020; Cho et al., 2023b), due to their limited generalization capabilities, find it challenging to perform effective evaluations. Hence, we adopt the recently popular MLLM-based question generation and answer framework (Cho et al., 2023a; Yarom et al., 2023; Hu et al., 2023b). As shown in Fig. 2 (b), given a evaluation prompt $p_{f,n}$, the video generation model $m$ produces a video $v$ that requires evaluation. Based on the semantic information in $p_{f,n}$, we utilize an LLM to generate the dimension-specific question set $Q$. Then, using the generated video $v$, we employ the MLLM to answer each question in $Q$, resulting in an answer set $A$. Finally, the evaluation

results $R$ are derived as a mapping from $A$. This can be formalized as:

$$(p_{f,n}) \xrightarrow{\text{m}} v, \quad (p_{f,n}) \xrightarrow{\text{LLM}} Q, \quad (Q, v) \xrightarrow{\text{MLLM}} A \to R. \tag{2}$$

Corresponding to the three evaluation dimensions mentioned above, our evaluation question set $Q$ comprises three categories: $Q_{\text{fid}}$, $Q_{\text{cov}}$, and $Q_{\text{coh}}$. Our three evaluation dimensions are represented as $R_{\text{fid}}$, $R_{\text{cov}}$, and $R_{\text{coh}}$. For some uncertain questions, during the process of deriving $A$ from $(Q, v)$, we observe that the MLLM tends to produce inconsistent answers across multiple repetitions for the same input. Moreover, the degree of uncertainty of a question directly influences the inconsistency of its answers (please refer to App. B.1 for more details). Thus, for the same $(Q, v)$ input, we instruct the MLLM to provide answers consecutively five times and use the proportion of a specific answer among these five as the final result, *i.e.*, $[(Q, v) \xrightarrow{\text{MLLM}} A]_{\times 5} \to R$. Fig. 3 illustrates the calculation process for each of our evaluation dimensions, as detailed below:

**Narrative element fidelity ($R_{\textbf{fid}}$).** To determine whether the generated video $v$ accurately conveys the narrative content of the prompt $p_{f,n}$, it is first essential to examine the generation of basic narrative elements represented by the scene and major objects in $p_{f,n}$ (Chatman & Chatman, 1980). Thus, in the step $(p_{f,n}) \xrightarrow{\text{LLM}} Q_{\text{fid}}$, we initially extract the following narrative elements based on the initial description in $p_{f,n}$: *scene category*, *scene attributes*, *object categories*, *object attributes*, *object actions*, and *initial layout of objects within the scene*. Elements missing in the prompt are ignored automatically. For each included element, we generate corresponding binary judgment questions $q_{\text{fid}}$, with answers $a_{\text{fid}}$ in [yes, no]. As depicted in Fig. 3, these questions form the set $Q_{\text{fid}} = \{q_{\text{fid}}^k\}_{k=1}^{N_{\text{fid}}}$, where the number of questions $N_{\text{fid}}$ is determined by the number of narrative elements in $p_{f,n}$.

Next, we perform the $[(Q_{\text{fid}}, v) \xrightarrow{\text{MLLM}} A_{\text{fid}}]_{\times 5} \to R_{\text{fid}}$ processing. For each question $q_{\text{fid}}^k$, the MLLM provides answers $\{a_{\text{fid}}^{k,t}\}_{t=1}^5$ through five iterations. We calculate the proportion of positive answers $a_{\text{pos}}^k$ (*i.e.*, *yes*) in the set $\{a_{\text{fid}}^{k,t}\}_{t=1}^5$ as the score $r_{\text{fid}}^k$ for that question. Finally, by computing the mean of all $r_{\text{fid}}^k$, we derive the final $R_{\text{fid}}$:

$$r_{\text{fid}}^k = \frac{1}{5} \sum_{t=1}^5 \delta(a_{\text{fid}}^{k,t}, a_{\text{pos}}^k), \quad R_{\text{fid}} = \frac{1}{N_{\text{fid}}} \sum_{k=1}^{N_{\text{fid}}} r_{\text{fid}}^k, \quad \text{where } \delta(x, y) = \begin{cases} 1, & \text{if } x = y \\ 0, & \text{otherwise} \end{cases}. \tag{3}$$

**Narrative unit coverage ($R_{\textbf{cov}}$).** For the narrative elements evaluated by $R_{\text{fid}}$, their temporal evolution forms the TNAs that encompass different narrative contents. Thus, $R_{\text{cov}}$ is primarily used to assess the coverage of the $n$ TNAs involved in the prompt $p_{f,n}$ by the generated video $v$. In the step $(p_{f,n}) \xrightarrow{\text{LLM}} Q_{\text{cov}}$, we first extract the TNA list corresponding to $p_{f,n}$. Then, we generate a judgment question $q_{\text{cov}}$ for each TNA regarding its existence, forming the question set $Q_{\text{cov}} = \{q_{\text{cov}}^k\}_{k=1}^{N_{\text{cov}}}$, where the number of questions $N_{\text{cov}}$ is determined by $n$, meaning the scope of the questions expands along with the expansion of TNAs. For the calculation of $R_{\text{cov}}$, we employ the same approach as Eq. 3.

**Narrative unit coherence ($R_{\textbf{coh}}$).** For the step $(p_{f,n}) \xrightarrow{\text{LLM}} Q_{\text{coh}}$, we first extract the TNA list corresponding to $p_{f,n}$. Then, we sequentially select pairs of adjacent TNA contents and generate judgment questions $q_{\text{coh}}$ regarding the existence of transitions between them. This forms the question set $Q_{\text{coh}} = \{q_{\text{coh}}^k\}_{k=1}^{N_{\text{coh}}}$, where $N_{\text{coh}}$ is also determined by $n$. Based on this question set, we apply the calculation method from Eq. 3 to obtain $R_{\text{coh}}'$. Additionally, considering that the existence of TNAs is a prerequisite for determining transitions between them, we introduce the proportion of TNA existence $\rho_{tna}$, which, along with $R_{\text{coh}}'$, determines the final $R_{\text{coh}}$:

$$\rho_{tna} = \frac{1}{N_{\text{cov}}} \sum_{k=1}^{N_{\text{cov}}} \Theta(r_{\text{cov}}^k - \tau_{\text{cov}}), \quad R_{\text{coh}} = \frac{1}{2}(R_{\text{coh}}' + \rho_{tna}), \quad \text{where } \Theta(x) = \begin{cases} 1, & \text{if } x > 0 \\ 0, & \text{otherwise} \end{cases}. \tag{4}$$

Here, we consider a TNA to exist if its corresponding $r_{\text{cov}}^k$ exceeds the threshold $\tau_{\text{cov}}$.

## 4 EXPERIMENTS

### 4.1 IMPLEMENTATION DETAILS.

**Evaluation models.** Our evaluation focuses on text-to-video models, a fundamental scenario in video generation (Li et al., 2019; Singer et al., 2022). First, our scope includes recently open-

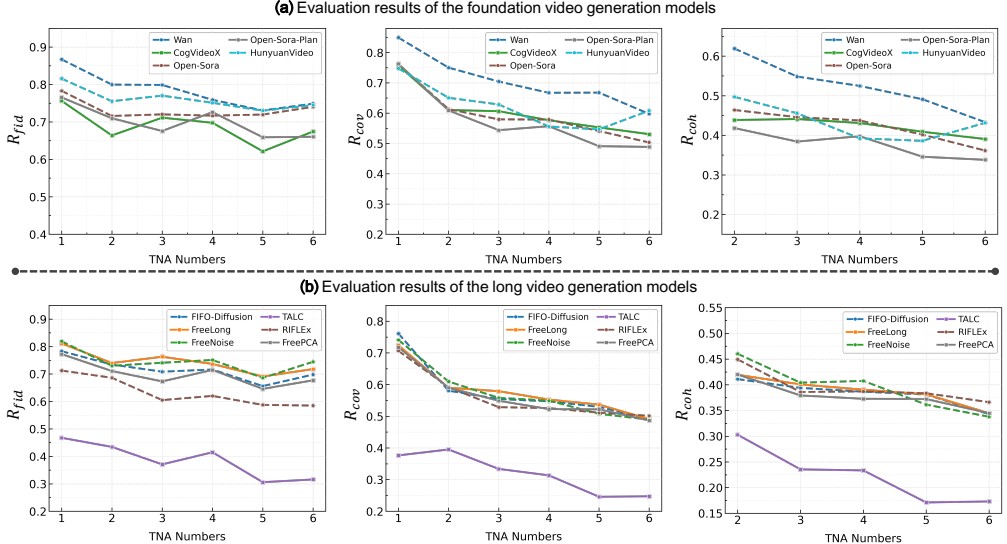

Figure 4: **Evaluation results across three evaluation dimensions.** Evaluated models include: **(a)** foundation video generation models and **(b)** long video generation models.

sourced long video generation models: TALC (Bansal et al., 2024b), FIFO-Diffusion (Kim et al., 2024), FreeNoise (Qiu et al., 2024), FreeLong (Lu et al., 2024), FreePCA (Tan et al., 2025), and RIFLEx (Zhao et al., 2025). Additionally, considering that many long-video generation models are derived from foundation video generation models, we find it necessary to include some of the latest mainstream open-source models. These include Wan2.1-14B (Wang et al., 2025), HunyuanVideo (Kong et al., 2024), CogVideoX1.5-5B (Yang et al., 2024b), Open-Sora 2.0 (Zheng et al., 2024b), and Open-Sora-Plan V1.3 (Lin et al., 2024). For the implementation details, please refer to App. C.

**Human annotation.** To analyze the alignment between our metric and human perception of narrative content expression, we perform human preference labeling on a large set of generated videos. Given a prompt $p_{f,n}$ and the models to be evaluated $\{m_j\}_{j=1}^{9}$, we randomly select two different models $(m_x, m_y)$, where $x \neq y$, to generate the corresponding video pairs $(v_x, v_y)$ for preference comparison. Corresponding to the three progressive dimensions in our metric, each video pair includes three questions (see App. D.1 for more details). Since $n = 1$ does not involve transition coherence between TNAs, we select test prompts within the range of $n \in [2, 6]$. Additionally, for each prompt, we select two video pairs, ultimately forming 600 pairs (*i.e.*, $1.8k$ questions) that require annotation. For each pair, we invite three human annotators. To ensure the correct understanding of the annotation task, we provide detailed training instructions to the annotators prior to the annotation process.

**Implementation settings.** In our prompt suite construction process, we utilized Qwen2.5-32B-Instruct (Yang et al., 2024a), which excels in text analysis and instruction-following capabilities, to extract scene and object elements from $200k$ text prompts. For the prompt generation pipeline, we chose GPT-4o (Hurst et al., 2024). For our evaluation metric, we employ the latest Qwen2.5-VL-72B-Instruct (Bai et al., 2025) as our MLLM. For the video input, we extract visual input by sampling 2 frames per second to feed into the MLLM. The threshold $\tau_{\text{cov}}$ is set to 0.3. All experiments were conducted on machines equipped with $8 \times$ H20 GPUs.

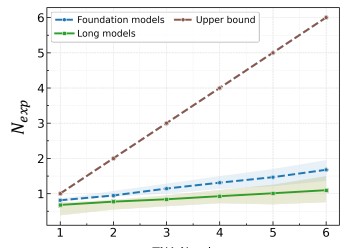

Figure 5: Evaluation on the number of TNA expressions $N_{\text{exp}}$.

## 4.2 EVALUATION RESULTS

Building on the NarrLV benchmark, we perform a series of evaluations (see App. D.2 for calculation details.) and distill four key observations regarding current video generation models.

**(i) Richer narrative semantics in text prompts weaken the model's representation of narrative units, while its ability to represent basic elements remains**

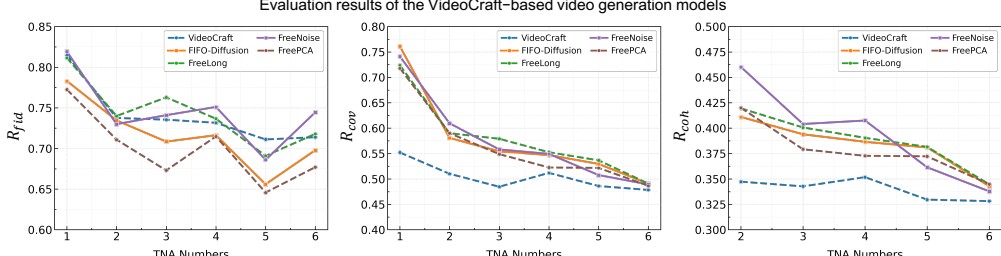

Figure 6: **Evaluation results across three evaluation dimensions.** Evaluated models include the foundation video generation model VideoCraft and the extended long video generation models (*i.e.*, FIFO-Diffusion, FreeLong, FreeNoise and FreePCA).

**relatively unaffected.** As shown in Fig. 4, we present the performance of foundation and long video generation models across three evaluation dimensions. As the number of TNAs increases, metrics for narrative units, namely $R_{cov}$ and $R_{coh}$, exhibit a noticeable downward trend, while the metric for narrative elements, $R_{fid}$, fluctuates within a small range. This suggests that even with text enriched in narrative content, the model is able to extract key elements for generation. However, constructing narrative content that evolves over time using these elements remains a challenge.

**(ii) Current models can only represent a very limited number of narrative units.** Considering that $R_{cov}$ reflects the average generation rate of TNAs, we introduce a new metric $N_{exp} = R_{cov} \times n$, which represents the number of TNAs that the model can effectively express. As shown in Fig. 5, with the increase in TNA numbers, $N_{exp}$ for both types of models shows a very slow increase, with the gap to the upper bound gradually widening. Therefore, when applying existing models, it is advisable that the number of TNAs contained in a given prompt does not exceed 2.

**(iii) The foundation model determines the narrative expression capability of the long video generation models derived from it.** Existing long video generation models are typically constructed by introducing specially designed modules onto the foundation model. For instance, FIFO-Diffusion, FreeLong, FreePCA and FreeNoise are all derived from VideoCraft (Chen et al., 2023; 2024a). Fig. 6 illustrates their performance. Interestingly, these models showcase similar capabilities in narrative elements (*i.e.*, $R_{fid}$). However, in terms of narrative unit expression capabilities (*i.e.*, $R_{cov}$ and $R_{coh}$), all long video models outperform VideoCraft, demonstrating the effectiveness of existing long video module designs. Nevertheless, the $R_{cov}$ and $R_{coh}$ of these long video models are quite similar, indicating that the capability of long video generation models largely depends on the foundation model employed. Although existing long video models perform less effectively than the latest foundation models (as shown in Fig. 4 and Fig. 5), these foundation models provide broad research opportunities for the advancement of long video generation.

**(iv) The impact of TNA change factors.** As shown in Tab. 1, we summarize the subsets corresponding to three factors (*i.e.*, $s_{att}$, $t_{act}$, $t_{att}$), and calculate the model's performance on the three evaluation dimensions. With respect to narrative element generation ($R_{fid}$), the model demonstrates superior average performance on the initial object action ($t_{act}$) compared to the other two factors ($s_{att}$, $t_{att}$). However, for narrative units ($R_{cov}$, $R_{coh}$), the model's performance is poorest along the object action factor ($t_{act}$). This indicates that the model excels in accurately generating a object action, but struggles with achieving diverse action variations.

Table 1: Comparison of model scores across three change factors under various metrics.

| Model | $R_{fid}$ | | | $R_{cov}$ | | | $R_{coh}$ | | |
|---|---|---|---|---|---|---|---|---|---|
| | $s_{att}$ | $t_{att}$ | $t_{act}$ | $s_{att}$ | $t_{att}$ | $t_{act}$ | $s_{att}$ | $t_{att}$ | $t_{act}$ |
| *Foundation video generation models* | | | | | | | | | |
| Wan | 74.9 | 77.8 | 82.5 | 68.8 | 72.7 | 70.3 | 50.1 | 52.4 | 54.5 |
| HunyuanVideo | 74.4 | 77.2 | 76.9 | 64.3 | 64.6 | 57.9 | 44.7 | 44.2 | 40.8 |
| CogVideoX | 67.3 | 69.9 | 69.1 | 62.9 | 60.2 | 58.6 | 44.5 | 38.9 | 43.1 |
| Open-Sora | 71.6 | 71.4 | 76.8 | 59.0 | 63.2 | 56.7 | 41.4 | 44.1 | 41.1 |
| Open-Sora-Plan | 68.5 | 67.8 | 73.6 | 59.3 | 60.6 | 52.7 | 38.9 | 39.0 | 35.2 |
| *Long video generation models* | | | | | | | | | |
| RIFLEx | 59.6 | 62.4 | 67.8 | 56.1 | 59.4 | 52.7 | 39.2 | 39.9 | 39.2 |
| FreeLong | 74.4 | 72.3 | 76.3 | 56.7 | 64.2 | 52.8 | 38.2 | 42.2 | 35.7 |
| FreeNoise | 77.6 | 71.5 | 74.5 | 58.5 | 63.0 | 51.2 | 40.7 | 43.1 | 34.4 |
| FreePCA | 69.6 | 67.8 | 72.3 | 55.7 | 60.5 | 53.2 | 37.1 | 40.4 | 35.8 |
| FIFO-Diffusion | 71.3 | 68.4 | 75.0 | 58.9 | 61.2 | 53.1 | 39.1 | 40.3 | 35.5 |
| TALC | 38.0 | 37.1 | 40.4 | 31.0 | 33.0 | 31.6 | 21.9 | 23.4 | 21.7 |
| **Mean** | **67.9** | **67.6** | **71.4** | **57.4** | **60.3** | **53.7** | **39.6** | **40.7** | **37.9** |

## 4.3 ADDITIONAL ANALYSIS

**Statistical analysis of our prompts suite.** Fig. 1 presents a statistical distribution of TNA numbers for our prompts compared to other representative benchmark prompts. Clearly, our prompt suite covers a broader and more uniform range of TNA numbers, facilitating a comprehensive evaluation of video generation models' narrative expression capability. Additionally, as shown in Fig. 7, we perform a word cloud analysis on 600 meticulously selected prompts. It is evident that words like *suddenly*, *next*, and *finally*, which pertain to the progression of narrative content, hold significant weight, aligning with our narrative-centric evaluation objectives.

Table 2: Comparison of metrics across different benchmarks. Consist-$n$/3 denotes the subset with $n$ consistent results out of three annotations.

| Metric | Consist-2/3 | | | Consist-3/3 | | |
|---|---|---|---|---|---|---|
| | $R_{\text{fid}}$ | $R_{\text{cov}}$ | $R_{\text{coh}}$ | $R_{\text{fid}}$ | $R_{\text{cov}}$ | $R_{\text{coh}}$ |
| VBench-2.0 | 0.33 | 0.32 | 0.28 | 0.31 | 0.27 | 0.29 |
| StoryEval | 0.41 | 0.51 | 0.51 | 0.55 | 0.55 | 0.56 |
| **Ours** | **0.63** | **0.67** | **0.67** | **0.81** | **0.80** | **0.79** |

Table 3: Ablation results on our metric. Consist-$n$/3 denotes the subset with $n$ consistent results out of three annotations.

| # | Variation | Consist-2/3 | | | Consist-3/3 | | |
|---|---|---|---|---|---|---|---|
| | | $R_{\text{fid}}$ | $R_{\text{cov}}$ | $R_{\text{coh}}$ | $R_{\text{fid}}$ | $R_{\text{cov}}$ | $R_{\text{coh}}$ |
| 1 | baseline | 0.63 | 0.67 | 0.67 | 0.81 | 0.80 | 0.79 |
| 2 | 1-response | 0.61 | 0.63 | 0.64 | 0.81 | 0.77 | 0.78 |
| 3 | 3-responses | 0.62 | 0.66 | 0.67 | 0.81 | 0.78 | 0.80 |
| 4 | adjust MLLM | 0.65 | 0.63 | 0.64 | 0.78 | 0.72 | 0.75 |

**Analysis of alignment with human judgments.** We use the video preference dataset annotated by three human participants, selecting data where two or three participants choose the same answers, which form subsets Consist-2/3 and Consist-3/3, and consider these annotations as groundtruth. Then, we analyze the evaluation accuracy of our metric and related metrics in alignment with this groundtruth (see App. D.1 for more details). The results in Tab. 2 indicate a high level of alignment between our metric and human perception, ensuring the reliability of the above evaluation conclusions. We compare our metric with the recent benchmarks involving narrative content evaluation, *i.e.*, VBench-2.0 (plot) (Zheng et al., 2025) and StoryEval (Wang et al., 2024a). Unlike VBench2.0, which uses video descriptions to make judg-

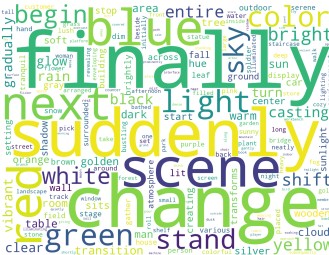

Figure 7: Word cloud analysis results of our prompt suite.

ments, and StoryEval, which requires the model to assess all narrative units at once, our progressive, question-driven approach demonstrates a significant performance advantage.

**Ablation analysis of metric design.** Tab. 3 (#1) presents the alignment accuracy of our metric with human judgments. Tab. 3 (#2) and Tab. 3 (#3) represent using MLLM to generate answers once and three times, respectively. As the frequency of responses increases, the accuracy correspondingly improves. However, when comparing Tab. 3 (#3) with Tab. 3 (#1), which uses 5-responses, accuracy shows signs of convergence. Hence, we choose the 5-responses approach. Finally, Tab. 3 (#4) denotes the replacement of the Qwen2.5-VL-72B with Qwen2.5-VL-32B. The results indicate that a reduction in MLLM capacity adversely affects accuracy.

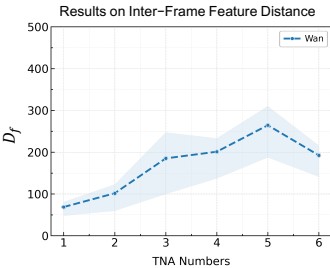

Figure 8: Analysis results on inter-frame feature distance $D_{\text{f}}$.

**Feature-level visualization analysis.** In addition to analyzing the generated result videos, we also aim to provide explanations from an intermediate feature level. Specifically, we introduce a metric $D_f$, defined as the average feature distance between consecutive frames. We obtain measurement results using the Wan2.1-14B under 6 TNAs and show the results in Fig. 8. Intuitively, an increase in the number of TNAs leads to a more information-rich video, resulting in a corresponding increase in inter-frame distances. However, due to the limited amount of information that can be conveyed within a unit of time, $D_f$ ultimately shows a converging trend. For implementation details, See App. D.3.

To intuitively understand the narrative expression capability of the model, we present in App. C.2 shows video generation results for prompts with different TNA counts and change factors.

## 5 CONCLUSION

To accommodate the pursuit of long video generation models for expressing rich narrative content over extended durations, we propose NarrLV, a novel benchmark dedicated to comprehensively assessing the narrative expressiveness of long video generation models. Inspired by the film narrative theory, we introduce a prompt suite with flexibly extendable narrative richness and an effective metric based on progressive narrative content expression. Consequently, we conduct extensive evaluations of existing long video generation models and the foundation generation models they typically depend on. Experimental results reveal the capability boundaries of these models across various narrative expression dimensions, providing valuable insights for further advancements. Moreover, our metric shows a high consistency with human judgments. We hope this reliable evaluation tool can facilitate future assessments of long video generation models.

## ACKNOWLEDGMENTS

This work is jointly supported by the National Science and Technology Major Project (No.2022ZD0116403) and the Strategic Priority Research Program of Chinese Academy of Sciences (No.XDA27010201).

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

APPENDIX

# A    MORE DETAILS ON OUR PROMPT SUITE

In this section, we will provide a comprehensive overview of the implementation details of our prompt suite.

## A.1    STATISTICAL ANALYSIS OF TNA NUMBERS IN EXISTING BENCHMARKS

Fig. 1 presents a statistical analysis of the number of TNAs in existing representative benchmarks such as VBench (Huang et al., 2024a), TC-Bench (Feng et al., 2024a), and StoryEval (Wang et al., 2024a). For StoryEval (Wang et al., 2024a), since it provides an event list corresponding to each prompt, and each event element in this list is a TNA of interest, we consider the length of this list as the number of TNAs contained in each prompt. However, for VBench (Huang et al., 2024a) and TC-Bench (Feng et al., 2024a), which lack corresponding structured representations, we follow recent evaluation and analysis studies based on LLMs (Li et al., 2024a;d;b;c), and employ GPT-4o (Hurst et al., 2024) to perform this text analysis task. Specifically, we employ the following instruction to analyze each text prompt to determine its corresponding number of TNAs.

---

**The prompt instruction for analyzing the number of TNAs in the text.**

As an expert in video narrative structure analysis, please analyze the given text based on the Temporal Narrative Atom (TNA). TNA is the minimal narrative unit in video generation that maintains continuous visual representation. It can be further understood through the following examples:
1. "A man is running" → TNA count is 1, as there is one continuous action.
2. "A person stands up from a chair and starts walking" → TNA count is 2, due to two actions ("stands up" → "walking").
3. "A room changes from bright to dim" → TNA count is 2, due to two environmental attributes ("bright" → "dim").

# Task Description
Given a text, please analyze the number of TNAs contained in this text.

# Example Demonstration
Input: A chameleon changes from brown to green
Output: 2

Based on the information provided above, please help me analyze the following text and only output the final result.
Input: **{User-provided information}**

---

Table A1: The major scene categories we study and their corresponding examples.

| Scene Category | Examples |
| --- | --- |
| Artificial Landscape | Garden, Fountain, Tree Nursery, Rice Field, Wheat Field, Hayfield, Cornfield, Vineyard, Lawn |
| Dining & Food Venue | Restaurant, Kitchen, Diner, Cafeteria, Fast Food Restaurant, Café, Dessert Shop, Food Court, Beer Hall |
| Commercial & Retail | Clothing Store, Bookstore, Jewelry Store, Gift Shop, Hardware Store, Pharmacy, Grocery Store, Pet Store, Shoe Store |
| Residential & Lodging | Apartment Building, Beach Villa, Cottage, Cabin, Mansion, Prefab Home, Treehouse, Mountain Lodge, Igloo |
| Transportation Hub | Airport, Raft, Bus Stop, Subway Station, Train Station, Parking Lot, Parking Garage, Highway, Port |

| Sports Venue | Soccer Field, Basketball Court, Baseball Field, Tennis Court, Golf Course, Race Track, Gymnasium, Volleyball Court, Boxing Ring |
|---|---|
| Industrial & Production Facility | Car Factory, Assembly Line, Repair Shop, Oil Rig, Industrial Zone, Energy Facility, Landfill, Warehouse, Assembly Line |
| Public Facility & Service | Fire Station, Police Station, Courthouse, Embassy, Post Office, School, Library, Lecture Hall, Science Museum |
| Arts & Entertainment | Art Gallery, Art Studio, Music Studio, Cinema, TV Studio, Nightclub, Carousel, Arcade, Amusement Park |
| Architectural Structure | Bridge, Arch, Corridor, Viaduct, Dam, Moat, Pavilion, Gazebo, Porch |
| Cultural & Religious Site | Church, Mosque, Temple, Synagogue, Mausoleum, Cemetery, Castle, Pagoda, Palace |
| Gaming & Virtual Environment | Game Scene, Sandbox Environment, Sci-Fi Scene, Animation Scene, VR/AR Enhanced Environment |
| Natural Geography | Forest, Rainforest, Desert, Beach, Coast, Glacier, Volcano, Canyon, Monolith |
| Other Special Scene | Military Base, Catacomb, Archaeological Dig, Battlefield, Trench |

## A.2 MORE DETAILS ON THE SCENE-OBJECT PAIR SET

Scenes and objects are the primary factors influencing TNA that we focus on, and they play a significant role in constructing our evaluation prompts. For $200k$ text prompts from VideoUFO (Wang & Yang, 2025) and DropletVideo (Zhang et al., 2025), we utilize Qwen2.5-32B-Instruct (Yang et al., 2024a) to extract the list of scenes and main objects corresponding to each text prompt. The prompt instruction used is as follows:

---

**The prompt instruction for analyzing the scene-object pair in the text.**

As an expert in video narrative structure analysis, please analyze the essential elements of the text description related to the video clip, to extract the corresponding scene and main objects set.

# Task Description
For a given text description about a video clip, you need to analyze its corresponding scene categories and list of main objects.
a. The scene category may appear directly in the text description. For texts without direct provision, you need to infer based on semantic content.
b. For the analysis of the main objects, please ignore some unimportant redundant information, such as subtitles in the video or OCR content on objects.

# Example Demonstration
Input: The video opens with a person standing in a dark room, surrounded by various digital screens displaying data and charts. The screens are colorful and dynamic, with different types of graphs and icons. The person appears to be in a virtual or augmented reality environment, as indicated by the holographic elements and the way the screens interact with the space. As the video progresses, the person turns around and looks at a smartphone, which is displaying a message that reads "LET'S KICKSTART THE FUTURE."
Output: {"Virtual Augmented Reality Environment": ["Person", "Smartphone", "Digital screens"]}

Based on the information provided above, please help me analyze the following text prompt strictly following the above JSON format and only outputting this JSON.
Input: **{User-provided information}**

---

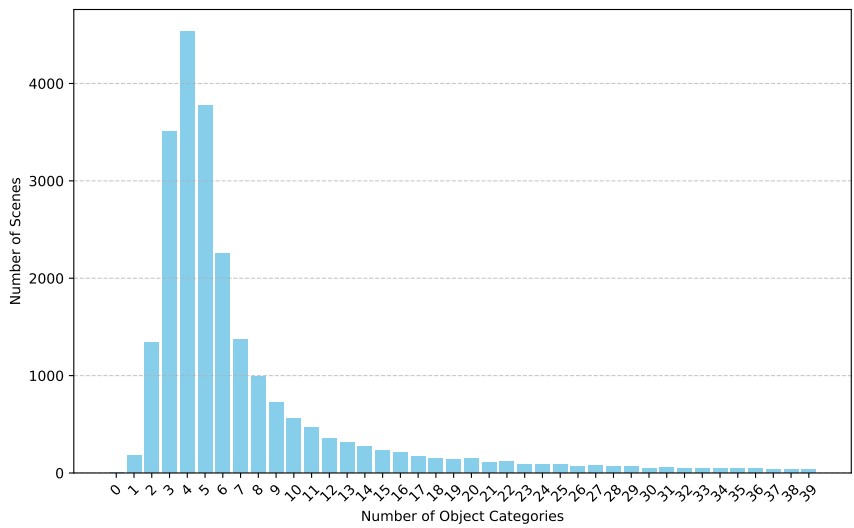

Figure A1: Statistical distribution of the number of object categories across different scenes.

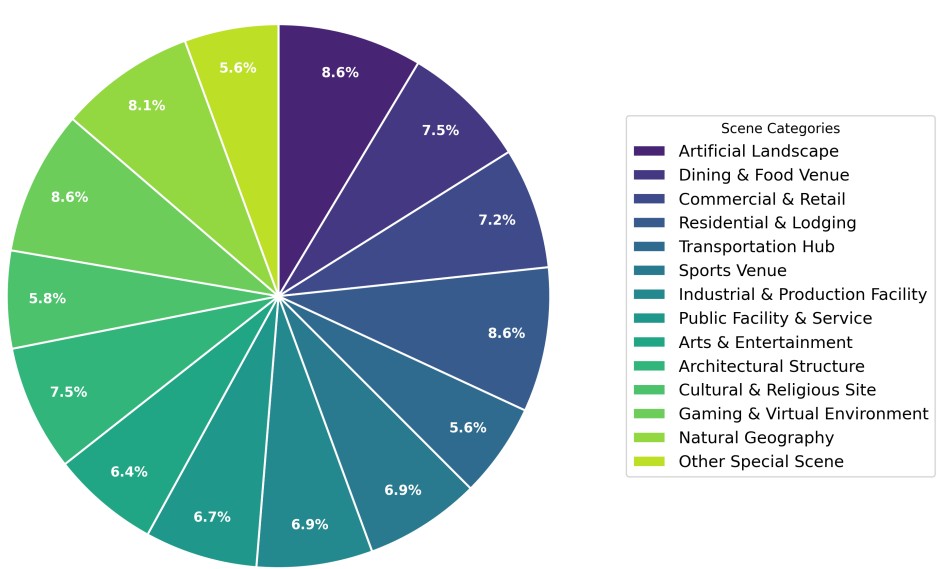

Figure A2: Statistical distribution of scene theme categories in our evaluation prompts (pie chart).

For each text prompt, we extract and obtain the corresponding scene and list of main objects. Subsequently, we merge objects within the same scene and record the frequency of occurrence for each object. Hence, each scene and its associated object list are considered as a scene-object pair $s_0$, forming our scene-object set $S_O = \{s_o\}$. After aggregation, we obtained $16k$ such $s_o$. We have compiled statistics on the number of object categories under different scenes, as shown in Fig. A1.

Due to the high computational cost of video generation, it is challenging to evaluate each specific scene comprehensively. Given the similarity among many scenes, we have classified these scenes into 14 major categories. Tab. A1 presents the names of these 14 categories along with examples of representative scenes. Considering that human-related scenes are more complex and diverse than natural scenes (Zhou et al., 2017; Feng et al., 2023; Yang et al., 2023), these categories are primarily constructed around human-related scenes. Although it is impractical to cover every individual scene, our evaluation prompts can encompass all of these 14 major scene categories, thus ensuring the diversity of scenes in our evaluation prompts. As shown in Fig. A2, we present the proportion of each scene theme category within our evaluation prompts as a pie chart. The balanced distribution across different scene categories indicates that our evaluation prompts are highly diverse.

### A.3 MORE DETAILS ON THE AUTOMATED PROMPT GENERATION PIPELINE

As illustrated in Fig. 2 (a) and Eq. 1, we utilize the sampled scene-object pair information $s_0$, the specified TNA number $n$, and the factor for TNA change $f$, to generate a specific evaluation prompt $p_{f,n}$ using GPT-4o. The prompt instruction we employ is as follows:

---

**The prompt instruction for evaluation prompt generation.**

As an expert in video narrative structure analysis, please analyze the given text based on the Temporal Narrative Atom (TNA). TNA is the minimal narrative unit in video generation that maintains continuous visual representation. It can be further understood through the following examples:
1. "A man is running" → TNA count is 1, as there is one continuous action.
2. "A person stands up from a chair and starts walking" → TNA count is 2, due to two actions ("stands up" → "walking").
3. "A room changes from bright to dim" → TNA count is 2, due to two environmental attributes ("bright" → "dim").

The reasons for TNA change in a video narrative are primarily:
1.Scene attribute changes
2.Object attribute changes
3.Object action changes

# Task Description
Your task is to generate a video segment description resulting in **{User-specified TNA count}** TNAs due to **{User-specified TNA change factor}** based on provided scene information and main objects:
1.Imagine an initial scene based on the provided scene information and objects. From this, describe the scene's overall attribute style (e.g., "overall grayish scene," "overall sunny") and position layout of main objects in the scene.
a. The number of provided objects is 1. Evaluate the reasonableness of including the object in the scene based on scene type. If unreasonable, the object may be omitted.
b. Extra objects may be introduced to meet the imagined scene requirements, but the total number of objects should not exceed 3.

2.Based on the initial scene, generate narrative content due to **{User-specified TNA change factor}** resulting in **{User-specified TNA count}** TNAs.
a. If the TNA change factor is "scene attribute changes," consider the potential attribute categories of scene and design a reasonable attribute evolution process.
b. If the TNA change factor is "object attribute changes," consider the potential attribute categories of object and design a reasonable attribute evolution process.

---

c. If the TNA change factor is "object action changes," consider the potential action categories of object and design a reasonable action evolution process.

3. Consolidate the initial scene description and subsequent TNA evolution into one text.
a. The final text should contain two parts: the initial scene and object layout description, followed by the TNA evolution description. Each part can be expressed in various forms.
b. Object layout description should introduce all potential objects, including those potentially involved in the TNA evolution description.
c. Object state and action description should be concise and clear.
d. The TNA count in the video segment text should match the specified count, and the type of TNA change should match the specified type.

# Example Demonstration
For generating video content descriptions due to **{User-specified TNA change factor}** with a TNA count of **{User-specified TNA count}**, here are reference examples:
    **<Examples of text for specific TNA count and TNA change factor >**
Based on the above prompt, please help generate the textual description for the following input.
Note: Only output the final description text without additional explanations.
Input: **{User-provided information}**

---

For the aforementioned **<Examples of text for specific TNA count and TNA change factor >**
, taking a TNA count of 3 and the change factor as "object action changes" as an example, we provide the following example:

> **The evaluation prompt examples for 3 TNAs with the "object action changes" factor.**
>
> 1. Example1:
> Input:
> - Scene: Undersea
> - Object: Coral
> Output: In the tranquil undersea world, vibrant corals spread out with a sea turtle hovering just above. Initially, the sea turtle slowly descends toward the corals. Then, the turtle stops and rests on a coral. Finally, the turtle starts swimming upwards.
>
> 2. Example2:
> Input:
> - Scene: Bench
> - Object: Person
> Output: In the gentle afternoon sunlight, a person sits quietly on a bench, reading a book. Then, the person puts the book away. Finally, the person stands up from the bench.

## B   MORE DETAILS ON OUR METRIC

In this section, we provide further implementation details of our evaluation metric.

### B.1   DISCUSSION ON MLLM ANSWERS TO UNCERTAIN QUESTIONS

Our evaluation metric computation employs a recently widely-adopted MLLM-based question generation and answering framework (Cho et al., 2023a; Yarom et al., 2023; Hu et al., 2023b), which leverages the powerful content understanding capabilities (Chu et al., 2025; Chen et al., 2025d) of MLLMs to perform robust evaluation (Chen et al., 2024b). For each question, we have an MLLM respond five times, and use the proportion of a specific answer among these five responses as the final outcome. We utilize this method because we have found that MLLMs tend to produce inconsistent answers across multiple repetitions for uncertain questions. Moreover, the degree of uncertainty of a question directly influences the inconsistency of its answers. As illustrated in Fig. A3, we present three questions concerning two video frame images, with each question requiring Qwen2.5-VL-72B

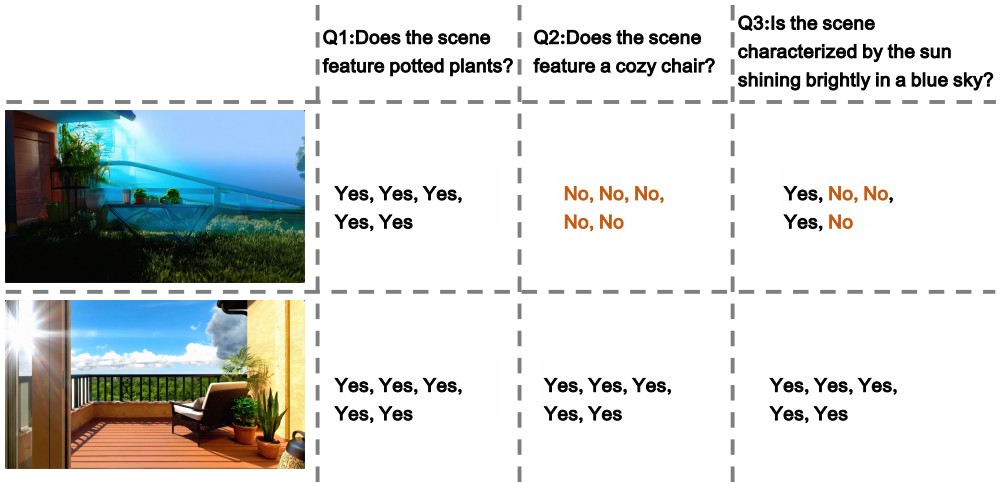

Figure A3: **An example illustrating the inconsistent responses of MLLM to uncertain questions.** For Q1 and Q2, MLLM provides consistent answers five times due to the clarity of judgment based on video frames. However, for Q3, the uncertainty present in the top image results in inconsistent responses from MLLM.

(Bai et al., 2025) to provide answers five times. The first two questions about the existence of objects yield consistent answers across all five responses from the MLLM due to the clear determination possible from the frame images. However, the third question concerning scene attributes shows inconsistency in answers based on top image, indicating uncertainty, whereas the bottom image, providing a clear basis for the question, results in completely consistent answers from the MLLM.

**Reproducibility of the proposed metrics.** As mentioned previously, MLLMs may produce inconsistent answers to the same question, and this variance is associated with the inherent ambiguity of the question. Therefore, we use the mean of multiple responses as the final answer. To assess the impact of this randomness (*i.e.*, non-zero temperature) on result reproducibility, we conducted a random error analysis under different sampling counts. Specifically, for a given sample count $n$, we draw $n$ responses for each evaluation and repeat this process three times. The mean absolute error between each pair among the three sets of results is then calculated and taken as the measure of random error. As shown in Tab. A2, the random error decreases monotonically with increasing $n$ and eventually stabilizes. When using five samples, the random error falls below 0.1%, indicating that our evaluation results are highly reproducible.

Table A2: Mean absolute error under different sample counts for various models.

| Model | 1 | 2 | 3 | 4 | 5 |
|---|---|---|---|---|---|
| Wan (Wang et al., 2025) | 0.0031 | 0.0025 | 0.0013 | 0.0009 | 0.0009 |
| CogVideoX (Yang et al., 2024b) | 0.0035 | 0.0019 | 0.0017 | 0.0008 | 0.0007 |
| HunyuanVideo (Kong et al., 2024) | 0.0024 | 0.0020 | 0.0013 | 0.0009 | 0.0008 |
| RIFLEx (Zhao et al., 2025) | 0.0044 | 0.0025 | 0.0013 | 0.0008 | 0.0008 |
| FreeNoise (Qiu et al., 2024) | 0.0038 | 0.0022 | 0.0014 | 0.0010 | 0.0009 |
| FIFO-Diffusion (Kim et al., 2024) | 0.0027 | 0.0024 | 0.0013 | 0.0009 | 0.0009 |

## B.2 More Details on the Implementation of Our Metric

The overall computation process of our progressive narrative-expressive evaluation metric is presented in Sec. 3.3. Here, we provide additional implementation details. Firstly, for the calculation of narrative element fidelity ($R_{\text{fid}}$), it is expected that the information on the scene and main objects of interest (Feng et al., 2025a) is well generated at the initial frame. Therefore, in the step $[(Q_{\text{fid}}, v) \xrightarrow{\text{MLLM}} A_{\text{fid}}]_{\times 5}$, we only use $v$ containing the initial frame image. Additionally, considering that the aesthetic

quality of the generated video affects narrative effectiveness at various levels (Arnheim, 1957; Doherty, 2013), we incorporate the aesthetic score of the initial video frame as a fixed offset, treating aesthetic questions as part of the question set and integrating it into the final metric calculation across the three metric dimensions. Specifically, we utilize the latest aesthetic evaluation model, Q-align (Wu et al., 2023), and map its aesthetic score to a 0 to 1 range. Since a dedicated aesthetic evaluation model is used for this aesthetic question, it needs to be answered only once by the model.

Additionally, given an evaluation prompt, we use GPT-4o to automatically generate corresponding questions. First, we utilize GPT-4o to organize the evaluation prompt into structured text. For the first evaluation dimension that focuses on scene and object elements, the structured text includes information on "Scene Type," "Main Object Category," "Initial Scene Attributes," and "Main Object Layout." For the second and third evaluation dimensions that focus on narrative unit information, the structured text contains list information derived from various TNA evolution states. The prompt instruction for implementing this operation is as follows:

---

**The prompt instruction for structured text extraction of the evaluation prompt.**

As an expert in video narrative structure analysis, please analyze the given text based on the Temporal Narrative Atom (TNA). TNA is the minimal narrative unit in video generation that maintains continuous visual representation. It can be further understood through the following examples:
1. "A man is running" → TNA count is 1, as there is one continuous action.
2. "A person stands up from a chair and starts walking" → TNA count is 2, due to two actions ("stands up" → "walking").
3. "A room changes from bright to dim" → TNA count is 2, due to two environmental attributes ("bright" → "dim").

The reasons for TNA change in a video narrative are primarily:
1. Scene attribute changes
2. Object attribute changes
3. Object action changes

# Task Description
You will be provided with a text list describing the TNA evolution process of video narrative content. Your task is to analyze the scene category and main objects, and from that, organize the evolution process concerning specific TNA change factors.

1. Regarding the text list, here are some additional introductions:
a. Each element describes only one TNA state, meaning the length of the list equals the TNA count of this entire video narrative content.
b. The elements in the list evolve sequentially over time; the first element in the list contains the initial scene description, also indicating the scene and main objects of the entire video narrative content, as well as the initial position layout information of the main objects in the scene.
c. The subsequent elements in the list focus primarily on specific TNA evolution descriptions.
d. Accompanying this list is the reason for TNA evolution for the entire timeline narrative content, being one of the three reasons mentioned above.

2. First, analyze the scene category, main object category, initial attributes of the scene, and initial position layout information of the main objects in the scene, mainly based on the initial scene description in the first element of the list.

3. Then, based on the subsequent list elements reflecting the TNA evolution situation and the reason for TNA change, extract the evolution states of the scene or object. The number of extracted evolution states should match the number of list elements.

4. The final output should follow a specific JSON format. Please refer to the format in the examples below for precise output.

---

# Example Demonstration
1. Example1:
Input: - Text narrative content: ["At the foot of the hill, lush vegetation thrives in the warm afternoon sunlight.", "Suddenly, rain begins to pour down, enveloping the entire scene."]
- TNA Change Reason: Scene Attribute Changes
- TNA Count: 2
Output: { "Scene Type": "foot of the hill", "Main Object Category": ["lush vegetation"], "Initial Scene Attributes": "warm afternoon sunlight", "Main Object Layout": "lush vegetation thrives at the foot of the hill.", "TNA Evolution States": ["warm afternoon sunlight", "rain enveloping the entire scene"] }

2. Example2:
Input:
- Text narrative content: ["On the balcony, there is a potted plant next to a watering can, and a man stands on the balcony gazing into the distance.", "The man begins to water the potted plant on the balcony.", "The man retreats back into the house, leaving the balcony."]
- TNA Change Reason: Object Action Changes
- TNA Count: 3
Output:{ "Scene Type": "balcony", "Main Object Category": ["potted plant", "watering can", "man"], "Initial Scene Attributes": null, "Main Object Layout": "a potted plant next to a watering can, and a man stands on the balcony gazing into the distance", "TNA Evolution States": ["the man stands on the balcony gazing into the distance", "the man waters the potted plant", "the man retreats back into the house"] }

Based on the above prompt, please assist me in extracting the structured text for the following input. Note: Only output the final structured text without additional explanations.
Input: **{User-provided information }**

Subsequently, based on this structured text, we utilize GPT-4o to generate corresponding judgment questions. The prompt instruction used is as follows:

## The prompt instruction for question generation.

As an expert in video narrative structure analysis, please analyze the given text based on the Temporal Narrative Atom (TNA). TNA is the minimal narrative unit in video generation that maintains continuous visual representation. It can be further understood through the following examples:
1. "A man is running" → TNA count is 1, as there is one continuous action.
2. "A person stands up from a chair and starts walking" → TNA count is 2, due to two actions ("stands up" → "walking").
3. "A room changes from bright to dim" → TNA count is 2, due to two environmental attributes ("bright" → "dim").

The reasons for TNA change in a video narrative are primarily:
1. Scene attribute changes
2. Object attribute changes
3. Object action changes

# Task Description
You will be provided with a JSON data analyzing the elements of the video narrative content. Your task is to generate a series of corresponding questions based on this JSON data. Each key-value pair corresponds to specific questions to be generated as follows:

1. "Scene Type": Corresponds to the scene location where the entire video narrative occurs. The question template to be generated is: "Does the scene take place in the xx(scene name)?"

2. "Main Object Category": Corresponds to the main objects involved in the entire video narrative. The question template to be generated is: "Does the scene feature xx(main object name)?"
a. Note: If multiple main objects are involved, a corresponding question needs to be generated for each object separately.

3. "Initial Scene Attributes": Corresponds to the initial attributes of the scene in the entire video narrative. The question template to be generated is: "Is the scene characterized by xx(initial scene attribute)?"

4. "Main Object Layout": Corresponds to the positional layout information of the main objects in the entire video narrative. The question template to be generated is: "Is the xx positioned as xxx(object and its positional layout information)?"
a. Note: If there is a positional layout relationship between multiple objects, a corresponding question needs to be generated for each layout relationship separately.

5. "TNA Evolution States": Corresponds to the information on the evolution of TNA states. This is a list data type, e.g., [TNA state 1, TNA state 2, xxx]. The method of generating questions and the corresponding template are as follows:
a. First, for each TNA state, generate a question separately, e.g., "Does the video contain any segments that xx(TNA state 1)?", "Does the video contain any segments that xx(TNA state 2)?",... This type of question matches the number of elements in the list.
b. Then, generate questions regarding the transition between adjacent TNA states. For adjacent TNA state 1 and 2, the question is: "Does the scene transition from xxx(TNA state 1) to xxx(TNA state 2) over time? To judge this question, it must first be determined that the scene exhibits xxx(TNA state 1) at a certain time and subsequently exhibits xxx(TNA state 2), with a clear transition process over time.

6. When generating specific questions, analyze the specific content to make adjustments. Feel free to adjust the templates to ensure the questions flow smoothly. The final output must follow a specific JSON format for structured output. Refer to the examples below for the exact format.

# Example Demonstration
1. Example1:
Input:
- TNA Element Information: { "Scene Type": "foot of the hill", "Main Object Category": ["lush vegetation"], "Initial Scene Attributes": "warm afternoon sunlight", "Main Object Layout": "lush vegetation thrives at the foot of the hill.", "TNA Evolution States": ["warm afternoon sunlight", "rain enveloping the entire scene"] }
- TNA Change Reason: Scene Attribute Changes
- TNA Count: 2
Output: { "Scene Type": ["Does the scene take place at the foot of the hill?"], "Main Object Category": ["Does the scene feature lush vegetation?"], "Initial Scene Attributes": ["Is the scene characterized by warm afternoon sunlight?"], "Main Object Layout": ["Does the lush vegetation thrive at the foot of the hill?"], "TNA Evolution States0": ["Does the video contain any segments showing warm afternoon sunlight?", "Does the video contain any segments where rain engulfs the entire scene?"], "TNA Evolution States1": ["Does the scene transition from warm afternoon sunlight to rain enveloping the entire scene over time? To judge this question, it must first be determined that the scene exhibits warm afternoon sunlight at a certain time and subsequently exhibits rain enveloping the entire scene, with a clear transition process over time."] }

2. Example2:
Input:
- TNA Element Information: { "Scene Type": "balcony", "Main Object Category": ["potted plant", "watering can", "man"], "Initial Scene Attributes": null, "Main Object Layout": "a potted plant next to a watering can, and a man stands on the balcony gazing into the distance", "TNA Evolution States": ["the man stands on the balcony gazing into the distance", "the man waters the potted plant", "the man retreats back into the house"] }

- TNA Change Reason: Object Action Changes
- TNA Count: 3
Output: { "Scene Type": ["Does the scene take place on a balcony?"], "Main Object Category": ["Does the scene feature a potted plant?", "Does the scene feature a watering can?", "Does the scene feature a man?"], "Initial Scene Attributes": null, "Main Object Layout": ["Is the potted plant positioned next to a watering can?", "Is the man standing on the balcony gazing into the distance?"], "TNA Evolution States0": ["Does the video contain any segments showing the man standing on the balcony and gazing into the distance?", "Does the video contain any segments showing the man watering the potted plant?", "Does the video contain any segments of the man retreating back into the house?"], "TNA Evolution States1": ["Does the man's action transition from standing on the balcony gazing into the distance to watering the potted plant over time? To judge this question, it must first be determined that the man is standing on the balcony gazing into the distance at a certain time and subsequently waters the potted plant, with a clear transition process over time.", "Does the man's action transition from watering the potted plant to retreating back into the house over time? To judge this question, it must first be determined that the man is watering the potted plant at a certain time and subsequently retreats back into the house, with a clear transition process over time."] }

Based on the above prompt, please assist me in extracting the structured text for the following input. Note: Only output the final structured text without additional explanations.
Input: **{User-provided information }**

## C MORE DETAILS ON OUR EVALUATED MODELS

In this section, we will provide further implementation details regarding our evaluated models and visualize some evaluation results.

### C.1 ADDITIONAL INTRODUCTION TO THE EVALUATED MODELS

We present the video duration, frame rate, and resolution information of the evaluated models in Tab. A5, with all data obtained based on the configuration of the official code. Due to the robust general-purpose generative capacity of foundation video generation models, most methodological improvements in video generation are built upon these pretrained models (Chen et al., 2025b;a; Wu et al., 2026; 2025a). Long video generation models follow the same paradigm. For TALC (Bansal et al., 2024a), it is implemented based on the foundation model ModelScopeT2V (Wang et al., 2023). For FreeLong (Lu et al., 2024), FreeNoise (Qiu et al., 2024), and FIFO-Diffusion (Kim et al., 2024), we adopted the official implementation based on VideoCraft2 (Chen et al., 2024a). For RIFLEx (Zhao et al., 2025), we opted for a twofold duration extension approach based on CogVideoX-5B (Yang et al., 2024b).

Recent advancements in long-video generation and multi-shot generation models have demonstrated substantial potential for producing rich narrative content. For example, to enable infinite-frame video generation, Skyreel-v2 (Chen et al., 2025c) introduces a novel diffusion model architecture and an effective multi-stage training scheme. Similarly, MAGI-1 (Teng et al., 2025) achieves chunk-level autoregressive generation through a sophisticated noise intensity design, facilitating flexible streaming long video synthesis. To enrich narrative content in generated long videos and support multi-shot scenarios, Captain Cinema (Xiao et al., 2025) proposes an innovative top-down keyframe planning and bottom-up video synthesis mechanism, enabling the generation of short movies. MovieDreamer (Zhao et al., 2024) adopts a similar hierarchical framework, leveraging the advantages of autoregressive and diffusion models to produce highly consistent keyframes, which are then extended to high-quality long videos. In contrast, VGOT (Zheng et al., 2024a) and MovieAgent (Wu et al., 2025b) utilize existing powerful content creation tools to construct intuitive and effective multi-agent frameworks. Specifically, VGOT employs a structured, step-by-step framework to address three core challenges: narrative fragmentation, visual inconsistency, and transition artifacts. MovieAgent develops an effective automatic film generation method based on a script and character bank by implementing systematic multi-agent chain-of-thought planning.

We attempt to include these models as part of our evaluation. However, because Captain Cinema (Xiao et al., 2025) and MovieDreamer (Zhao et al., 2024) are not open-source, we are unable to perform quantitative assessments on them. In addition, MovieAgent (Wu et al., 2025b) requires a specially constructed character bank as input, which is incompatible with the input format defined in our benchmark, rendering it unsuitable for evaluation within our framework. Taking these factors into account, we evaluate Skyreel-v2 (Chen et al., 2025c), MAGI-1 (Teng et al., 2025), and VGOT (Zheng et al., 2024a), and report the results in Tab. A7.

Furthermore, we analyze the computational efficiency of several representative foundation and long video generation models on an H20 GPU. Specifically, we evaluate the number of parameters in their key denoising networks (Params), the computational cost per forward operation (FLOPs), the time required for each forward operation (T), and the total number of forward steps needed for a complete generation process (Steps). The results in Tab. A6 indicate that recent foundation video generators, such as Wan2.1-14B and HunyuanVideo, possess extremely large parameter counts and correspondingly high computational costs. Additionally, current long-video models—including FreeLong, FreeNoise, and FIFO-Diffusion—are all built upon the same early foundation model (VideoCraft), resulting in identical parameter counts. However, their per-forward FLOPs differ due to each model employing a distinct strategy for long-video feature modeling.

Due to the substantial computational costs involved, existing long video generation models generally face challenges in generating significantly longer videos, and are typically limited to producing videos approximately 2 to 3 times the duration of their foundation counterparts. Unlike conventional video generation approaches, FIFO-Diffusion employs a unique denoising mechanism that enables the recursive generation of longer videos without a significant increase in computational cost. We extended its official default setting from 10 seconds to 60 seconds to analyze our benchmark's evaluation capability for minute-long videos. Tab. A3 shows that increasing the video duration led to an improvement in the model's narrative capability (the mean score increased from 0.57 to 0.59). We speculate that this is mainly because longer videos provide more space for content creation, thereby enabling the model to express narratives more effectively.

Table A3: Performance of FIFO-Diffusion on 10-second and 60-second video generation.

| Model | $R_{\text{fid}}$ | | | $R_{\text{cov}}$ | | | $R_{\text{coh}}$ | | | Mean |
|---|---|---|---|---|---|---|---|---|---|---|
| | $s_{\text{att}}$ | $t_{\text{att}}$ | $t_{\text{act}}$ | $s_{\text{att}}$ | $t_{\text{att}}$ | $t_{\text{act}}$ | $s_{\text{att}}$ | $t_{\text{att}}$ | $t_{\text{act}}$ | |
| FIFO-Diffusion (10s) | 0.75 | 0.74 | 0.79 | 0.59 | 0.61 | 0.53 | 0.39 | 0.41 | 0.35 | 0.57 |
| FIFO-Diffusion (60s) | 0.75 | 0.73 | 0.78 | 0.61 | 0.65 | 0.58 | 0.41 | 0.43 | 0.41 | 0.59 |

### C.2 VISUALIZATION OF EVALUATION RESULTS

To intuitively understand the narrative expression capability of the model, we present the video generation outcomes corresponding to prompts under different TNA counts and change factors, as shown in Fig. A6, Fig. A7 and Fig. A8. Intuitively, the increase in video length brings more challenges to the model (Feng et al., 2024b; 2025b;c), highlighting that there remains substantial room for improvement in the generative capabilities of existing long-video generation models.

Table A4: Analysis of answer consistency across different questions. Consist-$n$/3 denotes the subset with $n$ consistent answers out of three annotations.

| Metric | Consist-1/3 | Consist-2/3 | Consist-3/3 |
|---|---|---|---|
| $R_{\text{fid}}$ (Q1) | 81 | 361 | 158 |
| $R_{\text{cov}}$ (Q2) | 69 | 305 | 226 |
| $R_{\text{coh}}$ (Q3) | 73 | 309 | 218 |

## D MORE DETAILS ON THE EXPERIMENTS

In this section, we provide additional implementation details regarding our experiments.

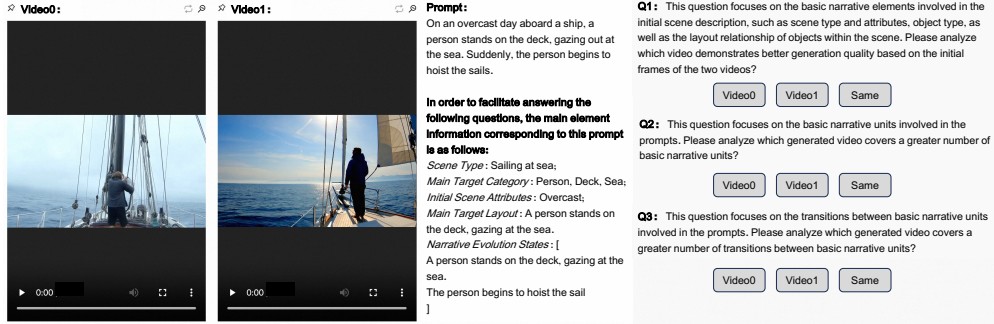

Figure A4: **Interface for human preference annotation.** From left to right, the interface includes a pair of videos to be compared, an evaluation prompt with corresponding structured element information, as well as three multiple-choice questions to be answered.

Table A5: Information on duration, frame Rate, and resolution of videos generated by our evaluation models.

| Model | Duration | Frame Rate | Resolution |
|---|---|---|---|
| Wan (Wang et al., 2025) | 5 s | 16 FPS | $1280 \times 720$ |
| HunyuanVideo (Kong et al., 2024) | 5 s | 24 FPS | $1280 \times 720$ |
| CogVideoX (Yang et al., 2024b) | 5 s | 16 FPS | $1360 \times 768$ |
| Open-Sora (Zheng et al., 2024b) | 5 s | 24 FPS | $336 \times 192$ |
| Open-Sora-Plan (Lin et al., 2024) | 5 s | 18 FPS | $640 \times 352$ |
| RIFLEx (Zhao et al., 2025) | 12 s | 8 FPS | $720 \times 480$ |
| FreeLong (Lu et al., 2024) | 12 s | 10 FPS | $512 \times 320$ |
| FreeNoise (Qiu et al., 2024) | 6 s | 10 FPS | $512 \times 320$ |
| FIFO-Diffusion (Kim et al., 2024) | 10 s | 10 FPS | $512 \times 320$ |
| TALC (Bansal et al., 2024b) | $2n$ s, if $n < 5$ 
 8 s, otherwise | 8 FPS | $256 \times 256$ |

### D.1 ANALYSIS OF METRIC ALIGNMENT WITH HUMANS

As introduced in Sec. 4.1, we conduct human preference annotations, which lay the foundation for subsequent analysis of the alignment between our metric and human perception. The human annotation interface is shown in Fig. A4. For each video pair, we provide the corresponding text prompt description. Additionally, to facilitate annotation, we also provide annotators with the structured information extracted from the evaluation prompts (see App. B.2). Based on this information, annotators are required to complete three judgment questions sequentially, which directly correspond to our three evaluation dimensions.

Our annotations were conducted by trained graduate-level researchers (Master's and PhD students) with backgrounds in computer vision. Each video pair was independently annotated by three annotators, and the process was supervised by two experts in video generation. For each annotation, annotators reviewed the full video and answered the corresponding questions, which took approximately 3–6 minutes per video depending on its length and complexity. Annotators were assigned a reasonable daily workload, and all annotation tasks were compensated at the standard research assistant rate.

Statistical analysis of the annotation results reveals that some video pairs have situations where three annotators choose three different answers. This means each option is selected a maximum of once, and we denote this subset as Consist-1/3. Additionally, we denote subsets with two or three participants selecting the same answers as Consist-2/3 and Consist-3/3. The sample sizes corresponding to these three subsets are shown in Tab. A4. Due to its poor consistency, we do not perform experimental analysis on the Consist-1/3 subset. For Consist-2/3 and Consist-3/3, we analyze the alignment between our metric and human preference. As indicated in Tab. 2, for the subset

with higher human consistency (Consist-3/3), our metric also shows better alignment with human preference.

To intuitively illustrate the effectiveness of our metric, we present qualitative examples in Fig. A5 showing evaluation results for different video pairs generated from the same prompt. These examples indicate that the differences in metric scores are well aligned with perceptual quality.

**IRB review.** Previous studies (Geirhos et al., 2021) have demonstrated that experiments solely involving interaction with computer systems (*i.e.*, screen and mouse) pose no risk to participants and therefore do not require IRB approval. Since our experiment follows the same procedure, we did not seek IRB review.

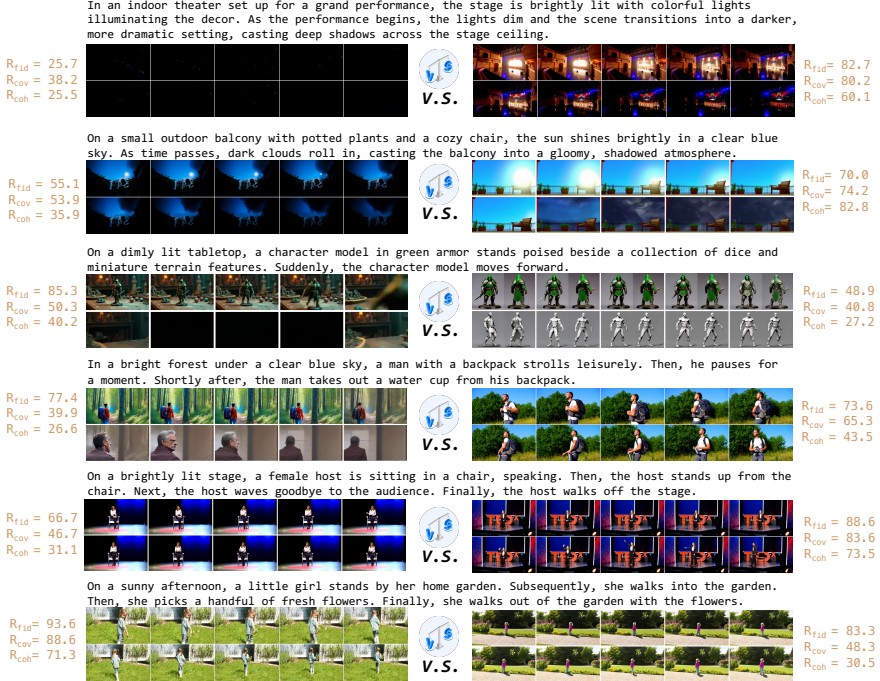

Figure A5: Evaluation prompts and corresponding generated video pairs. The results shown on both sides are calculated by our evaluation metric across three dimensions.

## D.2 MORE DETAILS ON THE EVALUATION RESULTS ANALYSIS.

Our evaluation results involve three different dimensions: TNA count $n \in [1, 6]$, TNA change factors $f \in [s_{att}, t_{act}, t_{att}]$, and our metric $R \in [R_{fid}, R_{cov}, R_{coh}]$. For each evaluation model, there are $6 \times 3 \times 3$ evaluation result data, denoted as $A$. We present all these evaluation results for foundation video generation models and long video generation models in Fig. A9 and Fig. A10, respectively. Although these results provide a detailed display of model performance across various dimensions, they do not readily facilitate the derivation of corresponding conclusions. The key observations presented in Sec. 4.2 are synthesized based on these evaluation results. Next, we introduce the specific process of this synthesis:

For **observation (i)**, we focus on the variations of the three metric indicators under different TNA counts. Thus, the results in Fig. 4 are obtained by averaging $A$ over the three TNA change factors. **Observation (ii)** focuses on the TNA expression quantity $N_{exp}$, which is constructed based on the $R_{cov}$ indicator and also averaged over the three TNA change factors. Furthermore, the results shown in Fig. 5 are statistically derived for both foundation video generation models and long video generation models. The solid lines represent the medians, and the shaded areas are determined by the 5th and 95th percentiles. **Observation (iii)** focuses on the variation in the three metric indicators for VideoCraft-based models (Chen et al., 2024a; Kim et al., 2024; Lu et al., 2024; Qiu et al., 2024) under different TNA counts. The calculation method in Fig. 6 is the same as that used in **observation (i)**.

**TNA=1:** In the calm, serene sea, a boat is gently floating with the horizon in the background.

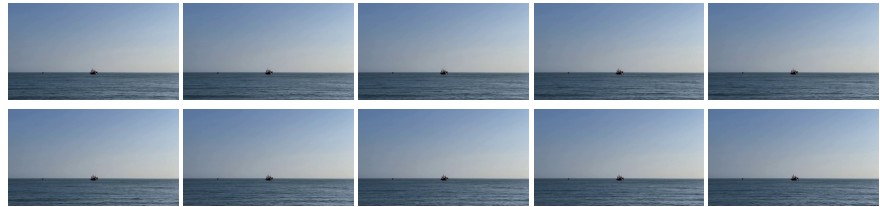

**TNA=2:** At the foot of the hill, lush vegetation thrives in the warm afternoon sunlight. Suddenly, rain begins to pour down, enveloping the entire scene.

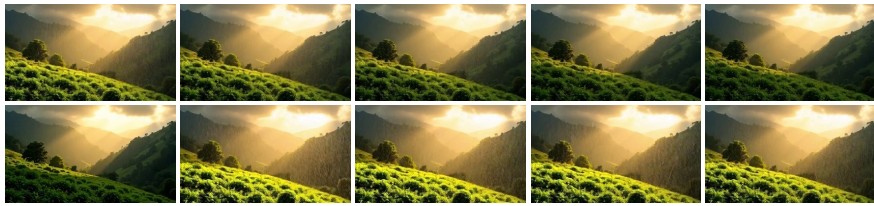

**TNA=3:** In a tranquil forest and grassland landscape, a helicopter hovers above the meadow, surrounded by lush greenery. Suddenly, a downpour occurs, enveloping the entire scene. Subsequently, the rain turns to snow, blanketing the forest and grassland with a thick layer of white.

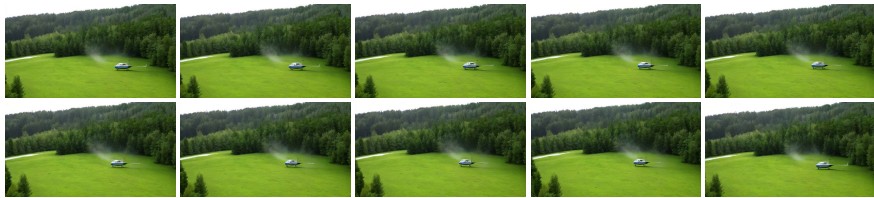

**TNA=4:** In a barren canyon with rugged, rocky walls and dry, dusty ground. Suddenly, a sandstorm approaches, reducing visibility as the entire scene becomes engulfed in swirling sand. Then, as the sandstorm passes, it begins to rain, and the ground darkens as it absorbs the water. Finally, the rain stops and the sun breaks through the clouds, leaving the canyon brighter with glistening wet rocks.

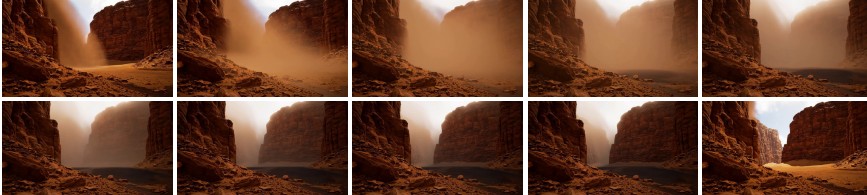

**TNA=5:** In an outdoor arid environment, a solitary man stands on a dusty, cracked earth under a bright, hot sun. Suddenly, clouds gather and cast shadows over the landscape, creating a cooler, dim atmosphere. Then, a sandstorm sweeps through, reducing visibility and covering the ground with a thin layer of sand. After the storm passes, a rare, gentle rain begins to fall, bringing a refreshing wetness to the scene. Finally, as the rain ceases, the sun sets, painting the sky with vibrant hues of orange and purple, casting long shadows over the transformed terrain.

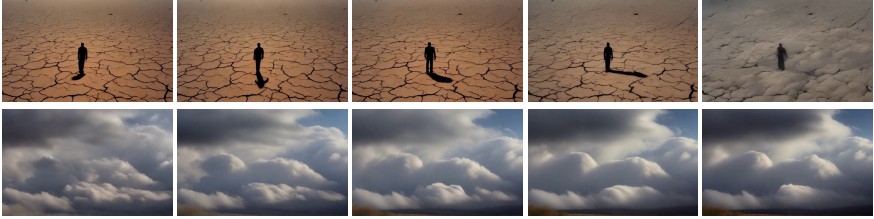

**TNA=6:** Beside a stretch of green grass and a gently flowing stream, tall trees and bushes grow under the bright sun. As time passes, the sunlight gradually diminishes, and the scene transitions into dusk, with a warm golden glow spreading across the area. Then, night falls, casting a deepening blue twilight. Soon, the sky becomes completely dark, and the entire scene is illuminated by the soft silver light of the moon. As dawn approaches, the sky and scenery are brushed with delicate pink hues. Finally, the sun rises fully, bringing back the vibrant colors and life to the grassy area by the stream.

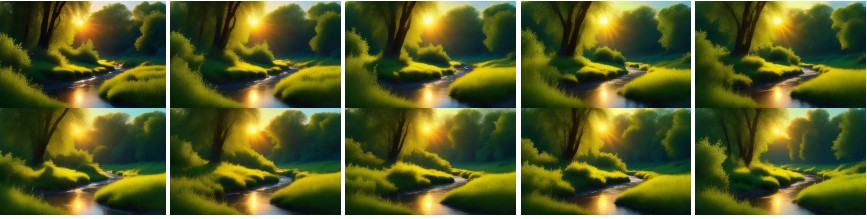

Figure A6: Evaluation prompts and corresponding generated videos under varying TNA numbers (1 to 6) induced by scene attribute change factors. The viewing order of video frames is from left to right, top to bottom.

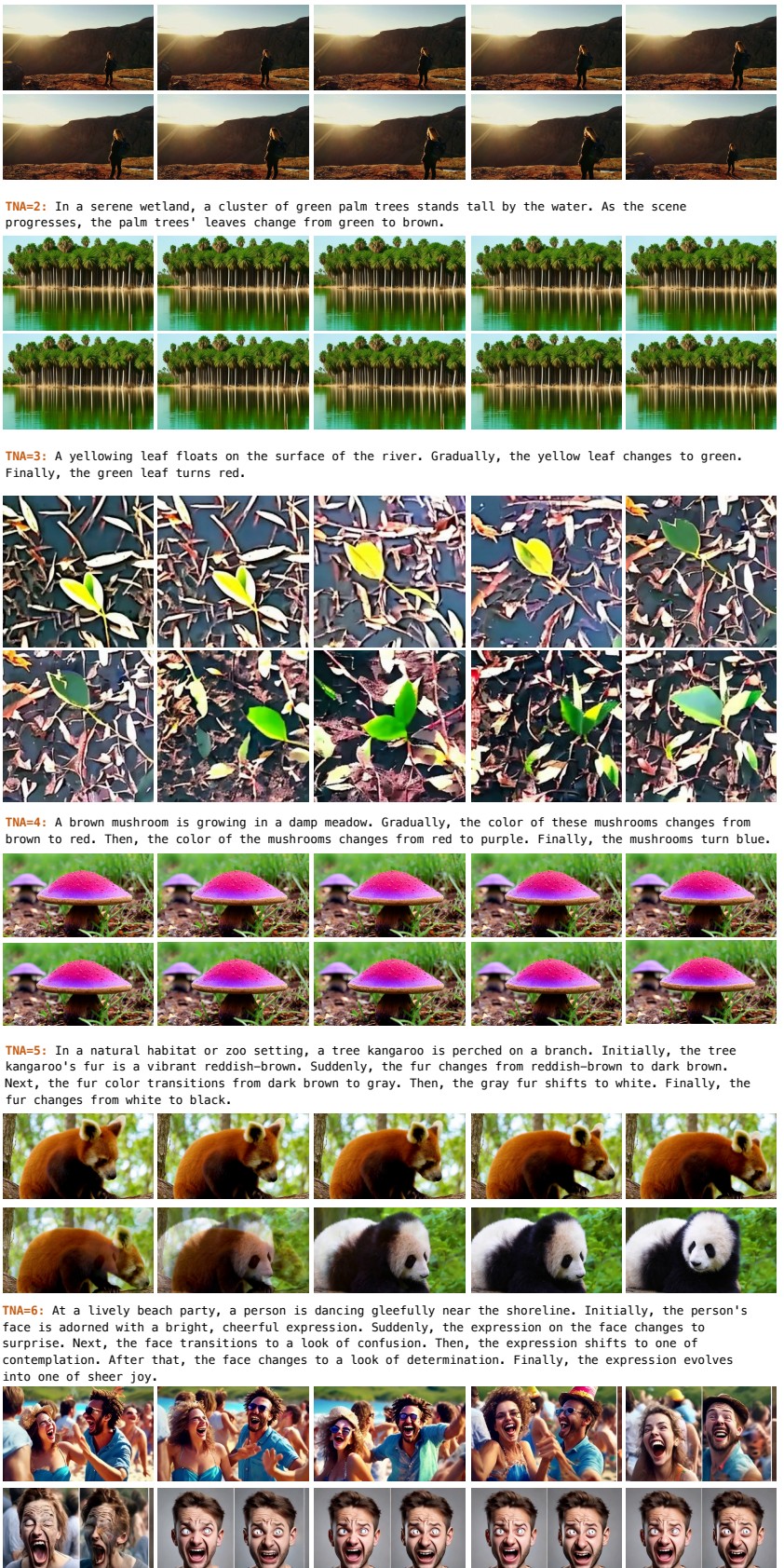

Figure A7: Evaluation prompts and corresponding generated videos under varying TNA numbers (1 to 6) induced by target attribute change factors. The viewing order of video frames is from left to right, top to bottom.

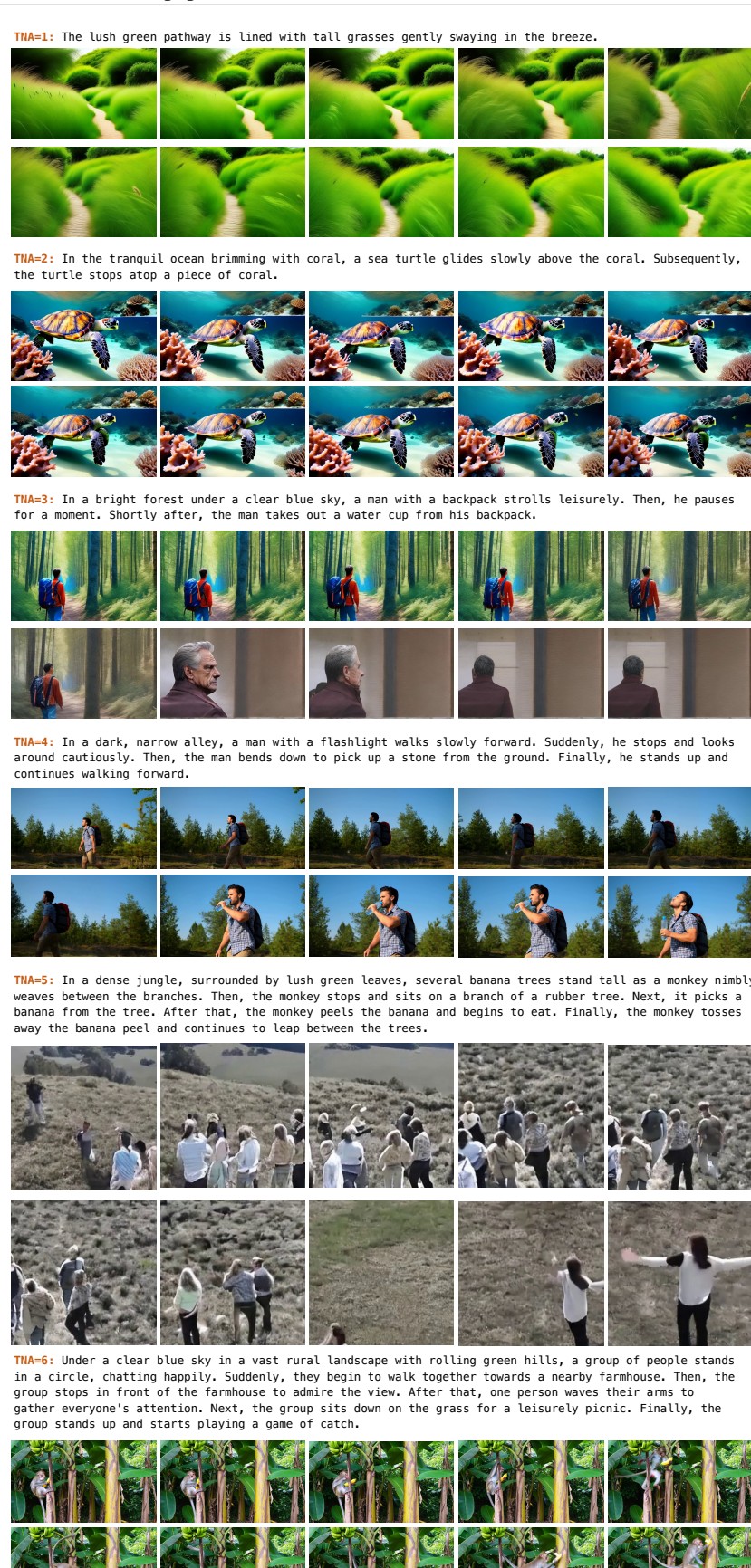

Figure A8: Evaluation prompts and corresponding generated videos under varying TNA numbers (1 to 6) induced by target action change factors. The viewing order of video frames is from left to right, top to bottom.

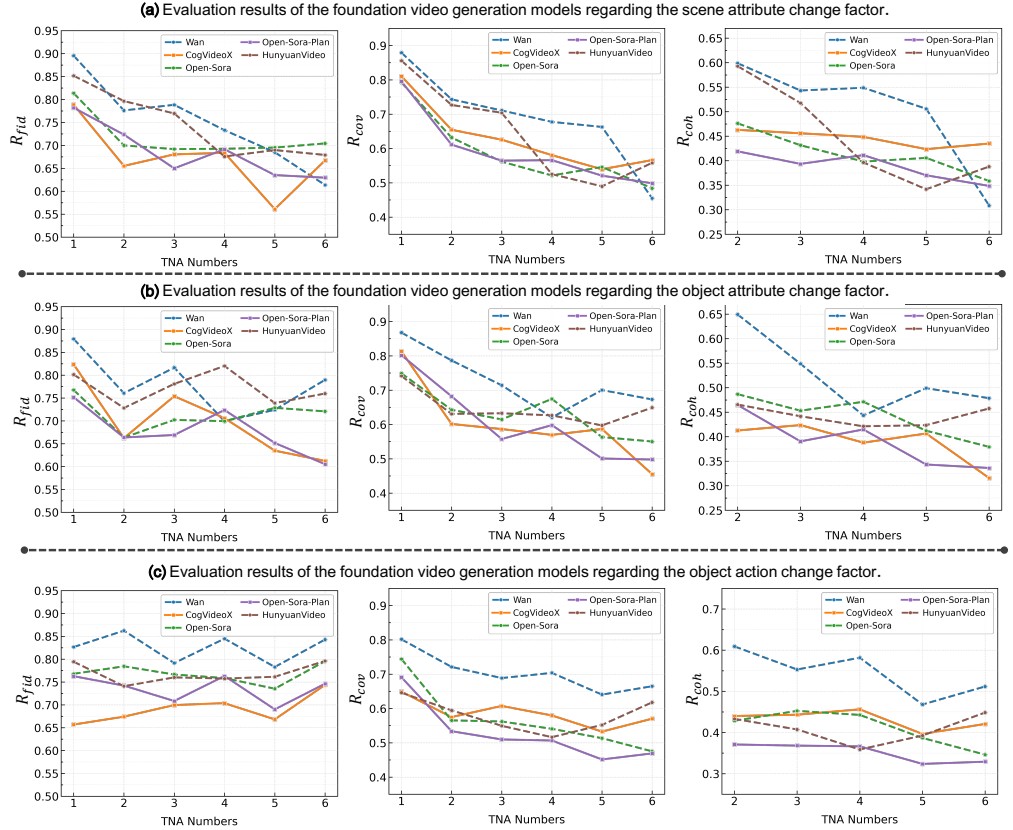

Figure A9: **Evaluation results across three evaluation dimensions and three TNA change factors.** The evaluated models comprise mainstream foundation video generation models (Wang et al., 2025; Yang et al., 2024b; Zheng et al., 2024b; Lin et al., 2024; Kong et al., 2024).

**Observation (iv)** focuses on the variation in the three metric indicators under different TNA change factors. Therefore, the results in Tab. 1 are obtained by averaging $A$ over the six TNA change ranges.

Table A6: Computational efficiency of different video generation models. Params, FLOPs, T, and Steps denote the number of parameters, computational cost per forward operation, time per forward operation, and total number of forward steps per generation, respectively.

| Model | Params (G) | FLOPs (T) | T (s) | Steps |
|---|---|---|---|---|
| Wan2.1-14B (Wang et al., 2025) | 14.3 | 904.9 | 111.6 | 50 |
| HunyuanVideo (Kong et al., 2024) | 12.8 | 351.6 | 131.8 | 50 |
| Open-Sora-Plan (Lin et al., 2024) | 2.8 | 100.8 | 3.5 | 100 |
| FreeLong (Lu et al., 2024) | 1.4 | 51.9 | 13.7 | 50 |
| FreeNoise (Qiu et al., 2024) | 1.4 | 23.5 | 6.6 | 50 |
| FIFO-Diffusion (Kim et al., 2024) | 1.4 | 5.9 | 1.1 | 800 |

## D.3 IMPLEMENTATION DETAILS OF FEATURE-LEVEL VISUALIZATION ANALYSIS

In addition to evaluating based on the final video generation results, we introduce the inter-frame feature average distance metric $D_f$ in Sec. 4.3, which facilitates analysis at the intermediate feature level. Specifically, for a given diffusion-based video generation model, we select the video latent space features $Z = \{z_i\}_{i=1}^{N_f}$ at the last denoising timestep, where $N_f$ denotes the number of video

Table A7: Performance of latest video generation models.

| Model | $R_{\text{fid}}$ | | | $R_{\text{cov}}$ | | | $R_{\text{coh}}$ | | |
|---|---|---|---|---|---|---|---|---|---|
| | $s_{\text{att}}$ | $t_{\text{att}}$ | $t_{\text{act}}$ | $s_{\text{att}}$ | $t_{\text{att}}$ | $t_{\text{act}}$ | $s_{\text{att}}$ | $t_{\text{att}}$ | $t_{\text{act}}$ |
| SkyReels-V2 | 72.7 | 69.0 | 75.0 | 65.5 | 68.9 | 59.9 | 46.9 | 51.5 | 43.1 |
| MAGI | 72.5 | 73.4 | 77.9 | 58.7 | 58.6 | 53.6 | 39.5 | 39.3 | 39.3 |
| VGOT | 83.0 | 79.1 | 79.0 | 70.1 | 72.0 | 52.0 | 51.5 | 56.2 | 35.2 |

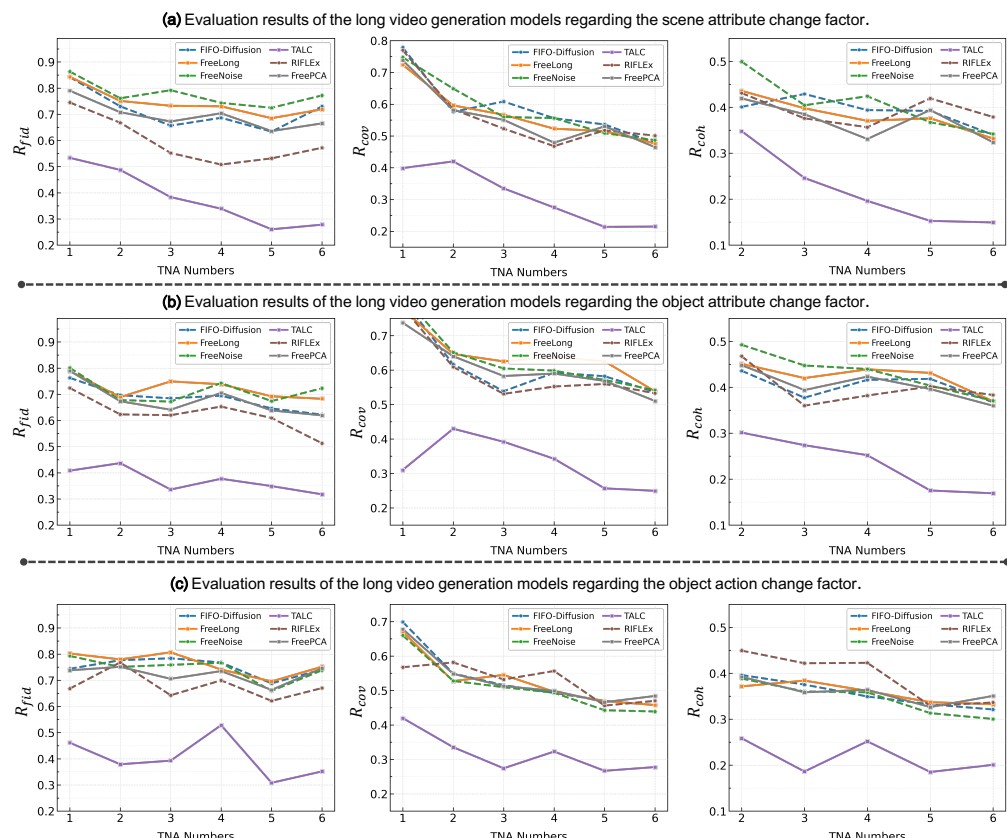

Figure A10: **Evaluation results across three evaluation dimensions and three TNA change factors.** The evaluated models comprise mainstream long video generation models (Kim et al., 2024; Qiu et al., 2024; Lu et al., 2024; Zhao et al., 2025; Bansal et al., 2024a).

frames. Then, $D_f$ is obtained through the following operation:

$$D_f = \frac{\sum_{i=1}^{N_f} \sum_{j=1}^{N_f} (z_i - z_j)^2}{(N_f)^2} \tag{A1}$$

This metric represents the average inter-frame feature distance for each video. For the results shown in Fig. 8, we select 15 prompts under each TNA for evaluation to ensure the reliability of the assessment outcomes. The solid lines represent the means, and the shaded areas are determined by the 30th and 70th percentiles.

# E   LIMITATIONS AND BROADER IMPACT

In this work, we propose a novel benchmark, NarrLV, which aims to comprehensively assess the narrative expressiveness of long video generation models. Currently, our evaluation primarily focuses

on open-source text-to-video models, which represent the fundamental task setting in the video generation domain. In the future, we intend to continually expand the scope of our evaluation models to include image-to-video models and cutting-edge open-source models. It is worth noting that, utilizing our established evaluation platform, we can directly test these models without requiring complex additional design.

Our NarrLV effectively reveals the narrative expressiveness of video generation models. Similar to many technologies centering around generative models, this work carries potential societal implications that warrant careful consideration (Katirai et al., 2024; Chen, 2023). Specifically, the models we assess with stronger narrative expression capabilities might facilitate the creation of deceptive or harmful video content. However, as advancements in video generation safety and regulatory technologies continue (He & Fang, 2024; Wang et al., 2024c; Dai et al., 2024), we believe these negative impacts will be progressively mitigated.

## F  USAGE OF LARGE LANGUAGE MODELS

Consistent with recent benchmarks (Zheng et al., 2025; Huang et al., 2024b; Wang et al., 2024a) in the field of video generation, we explore the integration of large language models (LLMs) into the design of benchmarks to enable automated evaluation. Specifically, existing LLMs are utilized as tools in both the construction of prompt generation pipelines and the implementation of MLLM-based question answering metrics. Detailed configurations of the employed LLMs are provided in the main text (please refer to Sec. 4.1). Moreover, we have employed GPT-4o to assist with the language polishing of this manuscript during its preparation.

