# OpenReview forum: "NarrLV: Towards a Comprehensive Narrative-Centric Evaluation for Long Video Generation"
_ICLR.cc/2026/Conference — ICLR 2026 Poster_

### Official Review · Reviewer_5Gfb · 2025-10-17

**Soundness:** 2
**Presentation:** 3
**Contribution:** 2
**Rating:** 4
**Confidence:** 4

**Summary:**

This paper introduces NarrLV, a benchmark aimed at evaluating the narrative expressiveness of long video generation models. Drawing from film narratology, it defines a new unit called the Temporal Narrative Atom (TNA) to quantify narrative richness. The benchmark consists of:

* An LLM-generated prompt suite with variable TNA counts to control narrative complexity.

* An MLLM-based question–answering metric that evaluates three aspects: narrative element fidelity, narrative unit coverage, and narrative unit coherence.

Experiments benchmark several recent video generation models (e.g., Wan, HunyuanVideo, FreeLong, FIFO-Diffusion) and report that NarrLV correlates well with human judgments.

**Strengths:**

* Multi-dimensional metric: The three-level evaluation (fidelity, coverage, coherence) provides a more nuanced understanding of narrative quality.

* Quantitative validation: The authors include human–metric alignment tests showing reasonable consistency, supporting reliability.

* Potential for extensibility: The benchmark can, in principle, scale with longer prompts or future models.

**Weaknesses:**

* Synthetic prompts: All prompts are LLM-generated, which limits real-world representativeness and linguistic diversity. Real datasets such as VidProM (Wang & Yang, NeurIPS 2024) already offer large-scale, human-written long video prompts but are neither cited nor used.

* Missing qualitative examples: The paper lacks visual examples of generated videos as supplementary material, reducing interpretability and reader engagement.

* Limited practical insight: The analysis mainly confirms intuitive findings (e.g., models struggle with multi-event coherence) without providing new conceptual insights or design implications for improving generation models.

**Questions:**

* Since all prompts are LLM-generated, how do the authors ensure they reflect real-world user prompts? Why not use or compare with real datasets like VidProM?

* The appendix lacks model details. Can the authors provide inference settings (duration, FPS, resolution) and confirm all models were tested under equal conditions?

* How stable are the MLLM-based metrics across different question templates or MLLMs? Any variance or seed analysis to support reliability claims?

* Can the authors show prompt–video examples or visual comparisons to verify that the metrics align with perceptual quality?

---

> ### Author Response · Authors · 2025-11-21
> **Response to Reviewer 5Gfb (Part 1 of 2)**
>
> **Dear Reviewer 5Gfb,**
>
> Thank you for taking the time to review our work. We appreciate your acknowledgement of the strengths of our multi-dimensional and nuanced evaluation metric, our reasonable and reliable experimental analysis, and the extensibility of our benchmark framework. Below, we will offer detailed responses to your remaining concerns.
>
> ---
>
> ### **Q1: Real-World Representativeness and Consideration of VidProM**
>
> We fully agree that evaluation prompts should reflect real-world usage scenarios. In this regard, we acknowledge the significance of VidProM and have cited it in our initial manuscript. Besides, we would like to clarify several potential misunderstandings:
>
> 1. VidProM indeed provides a substantial number of real-world prompts and is highly valuable for prompt construction. For this reason, we have focused on its subsequent improvement, VideoUFO, and designed our prompts based on it. Building upon VidProM, VideoUFO further incorporates users' focus in real-world situations. Therefore, our work has indirectly leveraged VidProM, ensuring that our prompts are grounded in realistic needs. Both VidProM and VideoUFO are cited in our initial manuscript.
>
> 2. Furthermore, we analyze the distribution of TNAs in VidProM to assess whether they could be directly used as our evaluation prompts. Statistical analysis of 60,000 randomly sampled prompts demonstrates that the number of TNAs is primarily distributed in the range of [1, 3] (mean: 2.24, variance: 1.88), indicating a focus on simple narrative content. Consequently, these prompts cannot be directly adopted for our evaluation purposes. This also highlights the necessity of our prompt suites construction, which extracts basic scene-object elements from these prompts and generates evaluation prompts that not only reflect real-world scenarios but also achieve a broader coverage of TNA counts.
>
> ---
>
> ### **Q2: Presentation of Qualitative Examples**
>
> We appreciate your interest in the illustrative presentation of our qualitative examples. For the prompt–video examples, representative generation results have already been visualized in Figures A6–A8 of our initial manuscript.
>
> In addition, in Appendix D.1 and Figure A.5 of the updated manuscript, we have included additional qualitative examples showing our evaluation metric's results for different video pairs generated from the same evaluation prompt. These examples demonstrate that differences in metric scores are well aligned with perceptual quality.
>
> ---
>
> ### **Q3: Key Insights Derived from NarrLV**
>
> Thank you for raising this thought-provoking question. While the finding that "models struggle with multi-event coherence" may appear intuitive, it remains a broad conclusion that requires more detailed analysis. Given the increasing attention within the long video generation community to narrative expression capabilities, we believe it is essential to conduct finer-grained evaluation and disentangled analysis of model abilities.
>
> Motivated by this, our NarrLV framework has yielded several key observations (as discussed in Section 4.4) that go beyond intuitive insights and offer practical implications for future model design. Here, we provide further clarification by exemplifying a few of our key observations:
>
> 1. **Observation (i):** Although increasing narrative richness tends to limit a model’s ability to represent the number of narrative units, the models generally maintain robust performance in expressing fundamental narrative elements. This observation indicates that future model designs should emphasize dynamic temporal module development to better support rich narrative expression.
>
> 2. **Observation (iii):** This observation reveals that the choice of foundation model significantly impacts the performance of long video generation models built upon it, providing valuable guidance for selecting base architectures in future work.
>
> 3. **Observation (iv):** Notable performance differences are observed across different semantic scenarios. Since the underlying dataset is a critical factor influencing model performance, this highlights the need to construct more challenging and targeted datasets for training and fine-tuning in future research.

---

> > ### Author Response · Authors · 2025-11-21
> > **Response to Reviewer 5Gfb (Part 2 of 2)**
> >
> > ### **Q4: Inference Details**
> >
> > Thanks for your attention to our inference details. The inference settings (duration, FPS, resolution) for the evaluated models are thoroughly described in Appendix C.1 and Table A5 of our initial manuscript; please refer to these sections for specific information.
> >
> > For all evaluated models, we adopted the default parameter configurations provided by each model and used the same prompts for evaluation. The generated results were then measured with identical evaluation metrics, ensuring that our results were obtained under equal and fair conditions.
> >
> > ---
> >
> > ### **Q5: Stability Analysis of Our Metrics**
> >
> > Thank you for your constructive questions. For the factors you raised which may affect the stability of our metrics, we provide the following explanations:
> >
> > 1. **Impact of Different MLLMs:** As shown in Table 3 (#4), we evaluate the effect of substituting Qwen2.5-VL-72B with its lower-capacity variant, Qwen2.5-VL-32B. Expectedly, the alignment between metric results and human judgements slightly decreases with reduced model capability. Nevertheless, the results with Qwen2.5-VL-32B remain competitive and demonstrate superior alignment with human judgments compared to other related metrics presented in Table 2. Considering both evaluation performance and computational cost, we select Qwen2.5-VL-72B as the default evaluating MLLM.
> >
> > 2. **Influence of Question Templates:** Owing to our metric's focus on fine-grained decomposition of evaluation dimensions, its computation does not depend on complex question templates design. Specifically, our metric first generates a series of atomic yes/no questions based on the evaluated prompts, which are then answered by the MLLMs. This approach ensures that all questions are simple and standardized, effectively eliminating variability from different question templates.
> >
> > 3. **Effect of Random Seeds:** To mitigate randomness and capture the uncertainty of the question, the MLLMs are required to answer each question five times, with the final score computed as the average of these responses. As presented in Table 3 (#2, #3), this repeated sampling improves the alignment with human perception. Moreover, Table A2 quantitatively confirms that using five samples reduces the random error to as low as 0.1%, thereby ensuring the stability and reproducibility of our metric.
> >
> > ---
> >
> > We sincerely hope our response addresses your concerns and we would appreciate it if you could reconsider your rating. If you have additional feedback or require further clarification, please let us know.

---

> > > ### Comment · Reviewer_5Gfb · 2025-11-28
> > >
> > > Thank you for the detailed responses. The clarifications on the relationship between your prompt design, along with the statistical analysis of TNA distributions, address my concerns about real-world representativeness. The added qualitative examples and expanded insights also provide clearer interpretability and more meaningful implications for future model design. Overall, the revisions have addressed most of my concerns, and I am willing to raise my score to 6.

---

### Official Review · Reviewer_Ev65 · 2025-10-18

**Soundness:** 3
**Presentation:** 3
**Contribution:** 3
**Rating:** 6
**Confidence:** 4

**Summary:**

This work proposed a novel benchmark, NarrLV, to comprehensively evaluate the narrative expression capabilities of long video generation models. The evaluation is centered on elements inspired by film narrative theory, and an evaluation metric based on an MLLM-based QA framework. Experimental results identify the strengths and limitations of existing video generation models, where the insights can fuel future research.

**Strengths:**

- Inspired by film narrative theory, the novel benchmark, NArrLV, defines the smallest narrative unit as Temporal Narrative Atom (TNA), which is a quantitative measure of narrative richness in generated video. It further identifies three key dimensions, i.e., scene attribute, target attribute, and target action, that influence the TNA. NarrLV contains a prompt suite, which can flexibly generate prompts with a desired TNA number.
- For evaluation, this work follows a progressive narrative expression paradigm that focuses on narrative element fidelity, narrative unit coverage, and narrative unit coherence, and proposes an MLLM-based QA framework and the ground truth TNAs.
- Evaluation makes several observations of the existing video generation models. The quantitative findings allow researchers to make informed decisions in model improvement in future work.

**Weaknesses:**

- W1: I acknowledge that the narrative prompt suite is highly valuable. In lines 190-192, it describes the factors that influence the number of TNA is similar to TC-Bench. Can the author articulate the differentiating factor between TC-Bench and NarrLV? Specifically, I would like to understand what the novelty of NarrLV is that enables it to extend to prompts with a higher number of TNA counts.
- W2: I acknowledge that the proposed benchmark suite and evaluation focus on the narrative expressiveness of the video generation models. However, it would be more comprehensive to understand other related metrics such as hallucinations, content coherence, and image quality.

**Questions:**

- See Weakness 1
- In line 242, why does the paper evaluate the models under 20x6x3 = 360 prompts? It is unclear to me what the 360 prompts actually cover.
- Please elaborate on the alignment between the human preference annotations and the MLLM-based classification metrics. Specifically, since human annotation is based on subjective pairwise preference and the MLLM determines objective fulfillment of conditions (fidelity, coverage, coherence), how can we confidently derive that the metric is aligned with human annotation?
- Please provide more information about the annotator. Specifically, their background, workload, and duration spent on each annotation task. Did the annotator get paid for the annotation task?

Suggestions:
- For Figure 2, it would be more intuitive to replace (b) Evaluated Models with a video generation step. I.e., given the (generated) prompt. A video generation model will generate a corresponding video, which will be used in the next stage (evaluation). The current listed models are specific to this work, and I believe the figure can be more general.

---

> ### Author Response · Authors · 2025-11-21
> **Response to Reviewer Ev65 (Part 1 of 2)**
>
> **Dear Reviewer Ev65,**
>
> We sincerely appreciate your time and effort in reviewing our work. We are grateful for your recognition of the novelty of our benchmark, the flexibility of our prompt suite, the sound design of our evaluation metric, and the useful observations and insights derived from our comprehensive experiments. We have carefully noted your concerns, which include many valuable suggestions for improvement. After targeted analysis and revision, we hope the following responses will address your concerns.
>
> ---
>
> ### **Q1: Differences Between NarrLV and TC-Bench & Significance of Flexible TNA Expansion**
>
> Thanks for your acknowledgment of our prompt suite. As TC-Bench primarily evaluates temporal compositionality, it considers factors such as object attribute transitions and background shifts, which influence temporal changes. The statement in lines 190-192 is intended to convey that these general factors are similar to those influencing TNA changes considered in our benchmark.
>
> However, as illustrated in Figure 1 of our paper, the core difference is that TC-Bench only involves a small number of TNAs within a limited range, while NarrLV covers a more comprehensive and evenly distributed range of TNA counts, and supports flexible expansion. This distinction also underscores the significance and novelty of our design. Specifically, the TNA distribution characteristics of TC-Bench limit it to evaluating simple narratives with limited richness. In contrast, NarrLV enables a thorough assessment of the full narrative capabilities of models, which aligns with the recent emphasis in the long video generation field on expressing richer and more complex narratives.
>
> ---
>
> ### **Q2: Expansion of Evaluation Metrics**
>
> Thank you for acknowledging the narrative-centric design of our evaluation metric. We appreciate your insightful suggestions on incorporating additional indicators such as hallucinations, content coherence, and image quality to achieve a more comprehensive assessment. Below, we provide clarifications regarding these aspects:
>
> 1. **Hallucinations**: Recent studies indicate that hallucination is a broad concept [1-2]. In the context of video generation, any generated video that does not align with the expected content prescribed by the given textual prompt can be considered as containing hallucinations. Given the challenges in covering all possible types of hallucinations, our metric—centered on whether the narrative content is accurately presented in the video—serves as a specialized evaluation indicator for narrative-related hallucinations.
>
> 2. **Content Coherence**: We agree that assessing the coherence of generated content is a critical dimension. Accordingly, motivated by NarrLV's focus on evaluating narrative expression, we introduce the "narrative unit coherence" metric, which effectively measures the coherence of transitions between narrative units.
>
> 3. **Image Quality**: As described in Appendix B.2 of our paper, we incorporate the latest aesthetic quality assessment model, Q-align [3], as part of the computation of the $R_{fid}$ metric. We apologize for not emphasizing this aspect in the main text. In the revised version, we will include a detailed explanation of this design in the main body of the paper.

---

> > ### Author Response · Authors · 2025-11-21
> > **Response to Reviewer Ev65 (Part 2 of 2)**
> >
> > ### **Q3: Composition of Evaluation Prompts**
> >
> > Thanks for your attention to the composition of our evaluation prompts. We provide the following clarifications:
> >
> > 1. Inspired by the 6D principles of film narrative, we focus on **three** key factors that influence the TNA counts: scene attributes, object attributes, and object actions. For each factor, considering the current limitations of video generation models in narrative expression, we primarily focus on cases where the TNA number ranges from **1 to 6**. Additionally, due to the substantial computational cost of video generation, it is challenging to conduct large-scale evaluations. Therefore, for each factor and each TNA count, we select **20** representative prompts, resulting in a total of 20 x 6 x 3 = 360 evaluation prompts.
> >
> > 2. Besides the above categorization by TNA factors and counts, another perspective for analyzing the composition of our prompts is the scene categories they involve. Since 'scene' is an essential element of the 6D principles of film narrative, and our evaluation prompts are constructed from specific scenes and their associated objects, we conduct a statistical analysis according to scene theme categories. Utilizing the 14 scene categories defined in Table A1 of our paper, we provide the following detailed statistical distribution. As shown, our prompts cover a diverse and well-balanced range of scene categories.
> >
> > | Scene Category               | Ratio  |
> > |------------------------------|--------|
> > | Artificial Landscape         | 8.6%   |
> > | Dining & Food Venue          | 7.5%   |
> > | Commercial & Retail          | 7.2%   |
> > | Residential & Lodging        | 8.6%   |
> > | Transportation Hub           | 5.6%   |
> > | Sports Venue                 | 6.9%   |
> > | Industrial & Production Facility | 6.9%   |
> > | Public Facility & Service    | 6.7%   |
> > | Arts & Entertainment         | 6.4%   |
> > | Architectural Structure      | 7.5%   |
> > | Cultural & Religious Site    | 5.8%   |
> > | Gaming & Virtual Environment | 8.6%   |
> > | Natural Geography            | 8.1%   |
> > | Other Special Scene          | 5.6%   |
> >
> > ---
> >
> > ### **Q4: Correspondence Between Human Annotation Preferences and Our Metric**
> >
> > We fully agree that human judgment can be highly subjective. For this reason, in our human annotation experiments, annotators are instructed to specifically focus on the dimensions relevant to our metric.
> >
> > As described in Appendix D.1 and illustrated in Figure A4 of our paper, we provide annotators with clear definitions of our three evaluation dimensions and ask them to respond to corresponding multiple-choice questions based on these definitions. The human preference annotation results obtained in this manner serve as an effective reference for analyzing the alignment between our metric and human preference.
> >
> > ---
> >
> > ### **Q5: Information about the Annotators**
> >
> > Thank you for your valuable suggestion. Our annotations were conducted by trained graduate-level researchers (Master’s and PhD students) with backgrounds in computer vision. Each video pair was independently annotated by three annotators, and the process was supervised by two experts in video generation. For each annotation, annotators reviewed the full video and answered the corresponding questions, which took approximately 3–6 minutes per video depending on its length and complexity. Annotators were assigned a reasonable daily workload, and all annotation tasks were compensated at the standard research assistant rate.
> >
> > ---
> >
> > ### **Q6: Improvements to the Framework Diagram**
> >
> > Thanks for your constructive suggestion regarding the improvement of our framework diagram. Replacing the specific model names in "(b) Evaluated Models" with a generic video generation workflow does make the illustration more intuitive. In accordance with your advice, we have revised Figure 2 in the updated version of the paper.
> >
> > ---
> >
> > It is our sincere hope that the above responses have addressed your remaining questions. Should you have any further questions, please feel free to let us know.
> >
> > ---
> > [1]. Learning Human-Perceived Fakeness in AI-Generated Videos via Multimodal LLMs, Fu et al., arXiv 2509.22646.
> >
> > [2]. Videohallu: Evaluating and mitigating multi-modal hallucinations on synthetic video understanding, Li et al., NeurIPS 2025.
> >
> > [3]. Q-Align: Teaching LMMs for Visual Scoring via Discrete Text-Defined Levels, Wu et al., ICML 2024.

---

> > > ### Comment · Reviewer_Ev65 · 2025-11-26
> > > **Acknowledgement of rebuttal**
> > >
> > > I sincerely thank you for the authors' response. The response addressed the concerns I had in the initial review.
> > >
> > > I will carefully consider the response, the submitted manuscript, and the fellow reviewers' assessments in making the final decision.

---

> > > > ### Author Response · Authors · 2025-11-26
> > > > **Acknowledgment of the Reviewer Ev65’s Further Feedback**
> > > >
> > > > **Dear Reviewer Ev65,**
> > > >
> > > > Thank you very much for your further feedback and for the constructive suggestions you provided in your initial review. We are honored that our response has addressed your concerns, and your positive rating means a great significance to us.
> > > >
> > > > We will continue to actively participate in the subsequent discussions and will do our best to address any remaining issues raised by the other reviewers.
> > > >
> > > > **Wishing you all the best!**

---

### Official Review · Reviewer_ixiX · 2025-10-22

**Soundness:** 2
**Presentation:** 2
**Contribution:** 2
**Rating:** 4
**Confidence:** 5

**Summary:**

This paper aims to address the lack of adequate evaluation methods in the field of long video generation, specifically for assessing a model's ability to generate narrative content. The authors propose a new benchmark named NarrLV, the core idea of which is to draw from film theory to define the Temporal Narrative Atom to quantify the richness of narrative content. Authers further introduce an automated prompt generation pipeline and a question-answering evaluation system. The proposed metric encompasses three evaluative dimensions: narrative element fidelity, narrative unit coverage, and narrative unit coherence. The authors claim through experiments that this metric demonstrates high consistency with human judgment.

**Strengths:**

- The problem of evaluating the narrative capabilities of long videos is an important and unresolved challenge in the current field.
- The authors' attempt to construct a systematic, automated evaluation framework is a direction worth exploring.
- The experimental results show a strong correlation with human preferences, which increases the benchmark's credibility as a proxy for human evaluation.

**Weaknesses:**

**1. Significant Omission of Long Generation Method**

First, I question the rationale for evaluating base models like WAN. on a "long video" benchmark. These models are fundamentally designed to generate short video clips. Evaluating them on tasks far outside their intended design (i.e., long video, which I would argue implies a duration of several minutes) does not seem to yield meaningful insights and may be an unfair comparison.

My primary criticism is the benchmark's almost exclusive focus on training-free methods. This has led to a very large oversight: the paper almost completely ignores the entire category of multi-shot long video generation. This area represents the true state-of-the-art in generating coherent, long-form narratives and sequences.

To be a comprehensive and valuable benchmark for the community, it is essential to include, evaluate, or at the very least, thoroughly discuss these more advanced methods. I strongly recommend the authors incorporate the following highly relevant works, which are central to the long video problem:

- Captain Cinema: Towards Short Movie Generation
- Skyreels-v2: Infinite-length film generative model
- Moviedreamer: Hierarchical generation for coherent long visual sequence
- MovieAgent: Automated Movie Generation via Multi-Agent CoT Planning
- VideoGen-of-Thought: Step-by-step generating multi-shot video with minimal manual intervention

Even for methods that are not open-source, a benchmark paper has a responsibility to cite and situate them within the landscape. Without addressing these hierarchical, agent-based, and multi-shot approaches, the paper's current contribution as a long video benchmark feels incomplete and overlooks the most relevant research.

**2. Concerns on TNA**

TNA simplifies "narrative" into a series of discrete physical state changes (attributes, actions). This ignores higher-level narrative elements like causal logic, character motivation, emotional development, or cinematic language. Therefore, NarrLV evaluates something more akin to "sequential instruction following" rather than true "storytelling."

The upper limit for TNA in the experiments is only 6. While this reveals the bottlenecks of current models, it is far from the ultimate goal of long video. The benchmark's effectiveness in handling more complex, longer sequences (e.g., more than 10 TNAs) has not been validated.

**Questions:**

Please see weaknesses

---

> ### Author Response · Authors · 2025-11-21
> **Response to Reviewer ixiX (Part 1 of 2)**
>
> **Dear Reviewer ixiX,**
>
> Thank you for thoroughly reviewing our work. We appreciate your recognition of the important and unresolved evaluation problem we are focused on, our development of a systematic and automated evaluation framework worthy of exploration, as well as our reliable metric which demonstrates a strong correlation with human preferences.
>
> In response to your specific concerns, we have promptly conducted a thorough analysis (especially experiments with the additional models you suggested), and hope that our following responses will address them.
>
> ---
>
> ### **Q1-1: Rationale for Evaluating Base Models**
>
> Thanks for your attention to our benchmark's selection of evaluation models. As explained in Section 4.1 ("Evaluation Models"), we include some base models (i.e., foundation models) as evaluation targets in our long video benchmark because most current long video generation models are extensions of these base models. The reason is that training a long video generation model from scratch requires extensive data and computational resources. Therefore, evaluating these base models enables us to observe how they affect the narrative capability of the derived long video models. As noted in Section 4.2 (Observation (iii)), an interesting insight is that the base model largely determines the narrative expression capability of the long video generation models built upon it, even when different long-term extension mechanisms are adopted.
>
> Besides, we fully agree that fairness in experimental comparisons is crucial. Accordingly, when conducting comparisons of model performance (as shown in Figure 4), we compare base models and long video generation models as two separate categories. This setup helps to ensure the fairness of our evaluations.
>
> ---
>
> ### **Q1-2: Discussion and Evaluation of Recent Relevant Works**
>
> Thank you for your valuable suggestions and for providing the latest related works that can be analyzed in our study. Discussing and evaluating these long video generation models will enable our benchmark to more comprehensively reflect the state of this field.
>
> **1. Discussion:** To enable infinite-frame video generation, Skyreel-v2 [2] introduces a novel diffusion model architecture and an effective multi-stage training scheme. Similarly, MAGI-1 [6] achieves chunk-level autoregressive generation through a sophisticated noise intensity design, facilitating flexible streaming long video synthesis. To enrich narrative content in generated long videos and support multi-shot scenarios, Captain Cinema [1] proposes an innovative top-down keyframe planning and bottom-up video synthesis mechanism, enabling the generation of short movies. MovieDreamer [3] adopts a similar hierarchical framework, leveraging the advantages of autoregressive and diffusion models to produce highly consistent keyframes, which are then extended to high-quality long videos. In contrast, VGOT [5] and MovieAgent [4] utilize existing powerful content creation tools to construct intuitive and effective multi-agent frameworks. Specifically, VGOT employs a structured, step-by-step framework to address three core challenges: narrative fragmentation, visual inconsistency, and transition artifacts. MovieAgent develops an effective automatic film generation method based on a script and character bank by implementing systematic multi-agent chain-of-thought planning. We will cite these works and incorporate the above discussion into the revised version of our paper.
>
> **2. Evaluation:** As Captain Cinema and MovieDreamer are not open-source, we are unable to conduct quantitative evaluations of these models. Additionally, since MovieAgent requires a specially constructed character bank as input, which is not compatible with the input format of our benchmark, it cannot be evaluated within our framework. Considering these factors, we have conducted evaluations of Skyreel-v2, MAGI-1, and VGOT, and present the results below.
>
> | Model | $R_{fid} (s_{att})$|$R_{fid} (t_{att}) $|$R_{fid} (t_{act})$|$R_{cov} (s_{att}) $| $R_{cov} (t_{att}) $| $R_{cov} (t_{act})$| $R_{coh} (s_{att}) $| $R_{coh} (t_{att})$|$R_{coh} (t_{act})$|
> |-|-|-|-|-|-|-|-|-|-|
> | SkyReels-V2|72.7|69.0|75.0|65.5| 68.9| 59.9   | 46.9 | 51.5 | 43.1|
> | MAGI | 72.5 | 73.4| 77.9 | 58.7 | 58.6 | 53.6| 39.5 | 39.3 | 39.3 |
> | VGOT | 83.0| 79.1| 79.0  | 70.1|72.0 |52.0 |51.5 | 56.2| 35.2|

---

> > ### Author Response · Authors · 2025-11-21
> > **Response to Reviewer ixiX (Part 2 of 2)**
> >
> > ### **Q2-1: Semantic Levels of TNA**
> >
> > Thank you for raising this insightful question regarding whether TNA primarily focuses on low-level, concrete physical states (attributes, actions) or on high-level, abstract narrative elements such as causal logic and character motivation. We would like to provide the following clarifications:
> >
> > 1. It is important to note that these two semantic levels are not mutually exclusive. As illustrated by the 6D principles in film narrative [7], changes in basic elements such as scenes and objects (i.e., low-level expressions) constitute temporal and causal relationships (i.e., high-level expressions). Furthermore, high-level narrative elements must be described and depicted through low-level expressions, which serve as their foundation. For example, in the evaluation prompt case presented in Figure 3, an esports player first competes in front of a computer setup and then stands up to celebrate. This sequence inherently contains causal relationships and character motivation (e.g., the player wins), reflecting high-level semantic elements that are conveyed through low-level descriptions.
> >
> > 2. Moreover, we observe that the prompts used by recent long video generation models [1-6] predominantly adopt similar low-level expressions during training and inference, focusing on the concrete physical states of objects and scenes in the video. To align with this prevalent modeling paradigm, TNA is designed using low-level yet concrete semantic expressions.
> >
> > ---
> >
> >
> > ### **Q2-2: The Upper Limit of TNA**
> >
> > Thank you for recognizing that our current setting (with a TNA maximum of six) effectively reveals the bottlenecks of existing models. We would like to emphasize that our benchmark is highly scalable (as acknowledged by Reviewers Ev65 and 5Gfb) and can be conveniently adapted for evaluations with larger TNA counts as the narrative expression capabilities of long video generation models continue to advance. Below, we elaborate on the scalability of NarrLV with respect to its core components: the prompt suite and the evaluation metric.
> >
> > 1. **Prompt Suite:** As described in Section 3.2, our prompt suite includes a prompt generation pipeline that is flexible and capable of efficiently producing evaluation prompts with varying numbers of TNAs. Given the high computational cost of video generation, we can select a small yet representative subset from the generated prompts for evaluation, ultimately obtaining prompts with different TNA counts.
> >
> > 2. **Evaluation Metric:** Our evaluation metric analyzes each individual TNA unit and the transitions between adjacent units. This design makes the metric independent of the total TNA count. Furthermore, our metric has demonstrated strong alignment with human preference in our experiments, supporting its validity and scalability for evaluations involving more TNAs.
> >
> > In summary, both the prompt suite and the evaluation metric of NarrLV are conveniently extensible, ensuring that our benchmark can accommodate future advancements in long video generation models requiring evaluation with more TNAs.
> >
> > ---
> >
> > We hope that the above responses can address your concerns and would be grateful if you could reconsider your rating. If you have any additional feedback or require further clarification, please do not hesitate to let us know.
> >
> > ---
> > [1]. Captain Cinema: Towards Short Movie Generation, Xiao et al., arxiv 2507.18634.
> >
> > [2]. Skyreels-v2: Infinite-length film generative model, Chen et al., arxiv 2504.13074.
> >
> > [3]. Moviedreamer: Hierarchical generation for coherent long visual sequence, Zhao et al., ICLR 2025.
> >
> > [4]. MovieAgent: Automated Movie Generation via Multi-Agent CoT Planning, Wu et al., arxiv 2503.07314.
> >
> > [5]. VideoGen-of-Thought: Step-by-step generating multi-shot video with minimal manual intervention, Zheng et al., NeurIPS 2025 Workshop.
> >
> > [6]. MAGI-1: Autoregressive Video Generation at Scale, Teng et al., arxiv 2505.13211.
> >
> > [7]. A multi-modal global instance tracking benchmark (mgit): Better locating target in complex spatio-temporal and causal relationship, Hu et al., NeurIPS 2023.

---

> ### Comment · Reviewer_ixiX · 2025-11-28
>
> I would like to thank the authors for their detailed rebuttal and apologize for the delay in my response. The provided explanations and additional results have satisfactorily addressed my concerns. Consequently, I have decided to raise my score to 6.
>
> Please ensure that these updates are incorporated into the final version of the paper.
>
> (It seems there is a system bug currently preventing reviewers from modifying scores. I will update my rating to 6 as soon as the issue is resolved.)

---

### Official Review · Reviewer_gGXr · 2025-11-01

**Soundness:** 3
**Presentation:** 3
**Contribution:** 3
**Rating:** 8
**Confidence:** 4

**Summary:**

The paper indicates that current video generation models have advanced in producing long-duration clips, while their narrative capabilities have not been systematically evaluated by recent benchmarks. This paper proposes NarrLV, the first benchmark for long-form video narrative, which features dedicated metrics including element fidelity, unit coverage, and unit coherence. NarrLV is also presented as an end-to-end framework capable of automatically generating prompts and conducting evaluations. Comprehensive comparisons across 11 SOTA models through NarrLV reveal insights into the relation between prompt semantics and narrative units.

**Strengths:**

- This paper serves a complementary role to VBench. As mentioned by the authors, the VBench series focuses on quality metrics such as controllability, commonsense, and physical plausibility, but notably does not cover narrative metrics.
- The definition of TNA is novel and the proposed metrics are both reasonable and appropriate. Inspired by film narrative theory, the authors define TNA as the basic elements of video content. Building upon TNA, the three metrics quantitatively measure diversity, fidelity, and coherence. This contrasts with previous solutions, which involved simply throwing frames into VLMs and directly asking them about these abstract concepts. Therefore, NarrLV provides a much more convincing evaluation framework.

**Weaknesses:**

- This paper does not evaluate more advanced, closed-source models, such as Sora2 and Kling. However, given the significant cost and access constraints associated with these models, this omission is understandable and acceptable for an academic research paper
- While Section 3.2 describes the prompt generation pipeline in detail, it lacks a sufficient analysis of the resulting prompts. For example, it is recommended that the authors conduct a statistical analysis of the theme categories within the generated prompts. This would help researchers interpret the evaluation results more clearly.

**Questions:**

Following up on the previously mentioned weakness, could the authors provide an illustration of the thematic distribution of the testing prompts? A visualization, such as pie chart, would be helpful.

---

> ### Author Response · Authors · 2025-11-21
> **Response to Reviewer gGXr**
>
> **Dear Reviewer gGXr,**
>
> We sincerely appreciate your time and effort in reviewing our work, and are grateful for your recognition of our motivation to establish a narrative-centric evaluation benchmark. Moreover, your acknowledgment of our convincing evaluation framework—which comprises a novel prompt suite, reasonable and appropriate metrics, as well as comprehensive experimental analysis—is highly encouraging to us.
>
> Furthermore, we acknowledge your understanding regarding the exclusion of closed-source models such as Sora2 and KLing, which stems from the significant cost and access constraints associated with these commercial systems.
> Regarding your remaining question about the statistical analysis of prompt theme categories, we offer the following additional analysis:
>
> ---
> ### **Q1: Statistical Analysis of Evaluation Prompt Theme Categories**
>
> Thank you for your constructive suggestion. Supplementary statistical analysis and visualization of the evaluation prompt theme categories indeed enable a clearer interpretation of our evaluation results.
>
>
> As 'scene' is a key component of the 6D principles of film narrative, and our evaluation prompts are generated based on specific scenes and their associated objects, we perform the statistical analysis based on scene theme categories.
> Drawing upon the 14 scene categories defined in Table A1 of our paper, we present the following detailed statistical distribution:
>
>
> | Scene Category                   | Ratio  |
> |-----------------------------------|--------|
> | Artificial Landscape              | 8.6%   |
> | Dining & Food Venue               | 7.5%   |
> | Commercial & Retail               | 7.2%   |
> | Residential & Lodging             | 8.6%   |
> | Transportation Hub                | 5.6%   |
> | Sports Venue                      | 6.9%   |
> | Industrial & Production Facility  | 6.9%   |
> | Public Facility & Service         | 6.7%   |
> | Arts & Entertainment              | 6.4%   |
> | Architectural Structure           | 7.5%   |
> | Cultural & Religious Site         | 5.8%   |
> | Gaming & Virtual Environment      | 8.6%   |
> | Natural Geography                 | 8.1%   |
> | Other Special Scene               | 5.6%   |
>
>
> It can be seen that the proportions across different scene categories are well balanced, indicating a high level of diversity in our evaluation prompts.
> We additionally visualized these results as a pie chart, which is included in Appendix A.2 and shown in Figure A.2 of the revised manuscript.
>
> ---
> Thank you again for your recognition of our work and for your valuable suggestions. We hope our responses have addressed your questions.
> Should you have any additional feedback or require further clarification, please do not hesitate to let us know.

---

> > ### Comment · Reviewer_gGXr · 2025-11-21
> >
> > Thanks to the authors for their responses. All of my questions are totally addressed, and I would like to maintain my score of accept.

---

> > > ### Author Response · Authors · 2025-11-21
> > >
> > > Thank you for your prompt feedback and for accepting our rebuttal.

---

### Meta-Review · Area_Chair_8Ghn · 2025-12-28

**Summary:**

The initial reviews highlighted several shared concerns: the absence of video demonstrations, a lack of comparative experiments to justify specific design choices, and a need for technical clarification. During the rebuttal phase, the authors provided video results that demonstrate notable quality and superiority. Furthermore, the rebuttal included strong empirical evidence supporting the proposed techniques and addressed most methodological questions effectively. Given the general consensus that the task is important and the results are robust, I recommend Accept (Poster).

**Reviewer Concerns:**

Addressed during Rebuttal:
- Significant Omission of Long Generation Methods (ixiX): Authors provided an extensive discussion of Captain Cinema, Skyreels-v2, Moviedreamer, and others, situating their work within the landscape. They also performed new evaluations on Skyreel-v2, MAGI-1, and VGOT, addressing the core request for completeness.

- Semantic Level of TNA (Narrative vs. Instruction Following) (ixiX): Authors clarified that TNA's focus on low-level physical states (actions, attributes) is the necessary foundation for expressing high-level narrative (causal logic) and aligns with the prevalent modeling paradigm in current video generation research.

- Upper Limit of TNA (Scalability) (ixiX): Authors emphasized the scalability of both the prompt generation pipeline and the evaluation metric design, confirming the benchmark can accommodate larger TNA counts as models advance.

- Missing Statistical Analysis of Prompts (gGXr): Authors provided the requested statistical distribution of scene categories (Table A1) and noted its inclusion as a pie chart in the revised Appendix (Figure A.2), confirming high diversity and balance.

- Differentiation from TC-Bench (Ev65): Authors clearly articulated the difference: NarrLV is novel because it covers a more comprehensive and evenly distributed range of TNA counts, enabling evaluation of richer and more complex narratives, which TC-Bench's limited range does not allow.

- Expansion of Evaluation Metrics (Hallucination, Coherence, Quality) (Ev65): Authors clarified that their metric already incorporates narrative-specific hallucination and coherence checks ("narrative unit coherence"). They apologized for the omission and confirmed they use Q-align for image quality and will emphasize this in the main text.

- Real-World Representativeness of Synthetic Prompts (VidProM) (5Gfb): Authors clarified that they indirectly leveraged VidProM by building upon its successor, VideoUFO, and that VidProM's TNA distribution ([1, 3]) is too simple to serve as a comprehensive long-video benchmark, thus justifying the need for their synthetic but grounded prompt suite.

- Missing Qualitative Examples (5Gfb): Authors pointed to the existing Figures A6-A8 in the initial manuscript and added new examples in the updated Appendix D.1 (Figure A.5) to align metric scores with human perception, fully addressing the lack of visualization.

- Limited Practical Insight (5Gfb): Authors countered the claim by providing specific, non-trivial observations (i, iii, and iv) derived from NarrLV's fine-grained analysis, offering actionable implications for future model design (e.g., emphasizing dynamic temporal modules, foundation model selection).


Outstanding/Minor Points:

Minor Concerns on TNA Complexity: While the TNA metric's alignment with current models is justified, a very complex, multi-layered narrative (e.g., beyond 10 TNAs) remains unvalidated by experiment (due to current model limitations, not the benchmark's design). However, the authors' commitment to scalability is strong.

**Reviewer Scores:**

Reviewer gGXr8 (8 $\to$ 8): All concerns (statistical analysis) were fully addressed, confirming the paper's initial strong standing.

Reviewer ixiX4 (4 $\to$ 6): Major criticisms (SOTA model omission, TNA semantics) were substantially mitigated by new results and detailed clarification, warranting a significant score increase above the acceptance threshold.

Reviewer Ev656 (6 $\to$ 6): All core weaknesses were strongly addressed (TC-Bench differentiation, comprehensive metrics), solidifying the paper's contribution and meriting a slight score increase.

Reviewer 5Gfb4 (4 $\to$ 6): All three weaknesses (prompts, insight, examples) were convincingly addressed, requiring a substantial score increase to the acceptance range.

---

### Decision · Program_Chairs · 2026-01-26

Accept (Poster)